# GAMR: A Guided Attention Model for (visual) Reasoning

**Mohit Vaishnav**[1,2,3]   **Thomas Serre**[1,2]

[1] Artificial and Natural Intelligence Toulouse Institute, Université de Toulouse, France
[2] Carney Institute for Brain Science, Dpt. of Cognitive Linguistic & Psychological Sciences
Brown University, Providence, RI 02912
[3] Centre de Recherche Cerveau et Cognition, CNRS, Université de Toulouse, France
`mohit.vaishnav@univ-toulouse.fr`

## Abstract

Humans continue to outperform modern AI systems in their ability to flexibly parse and understand complex visual scenes. Here, we present a novel module for visual reasoning, the Guided Attention Model for (visual) Reasoning (*GAMR*), which instantiates an active vision theory – positing that the brain solves complex visual reasoning problems dynamically – via sequences of attention shifts to select and route task-relevant visual information into memory. Experiments on an array of visual reasoning tasks and datasets demonstrate GAMR's ability to learn visual routines in a robust and sample-efficient manner. In addition, GAMR is shown to be capable of zero-shot generalization on completely novel reasoning tasks. Overall, our work provides computational support for cognitive theories that postulate the need for a critical interplay between attention and memory to dynamically maintain and manipulate task-relevant visual information to solve complex visual reasoning tasks.

## 1  Introduction

Abstract reasoning refers to our ability to analyze information and discover rules to solve arbitrary tasks, and it is fundamental to general intelligence in human and non-human animals (Gentner & Markman, 1997; Lovett & Forbus, 2017). It is considered a critical component for the development of artificial intelligence (AI) systems and has rapidly started to gain attention. A growing body of literature suggests that current neural architectures exhibit significant limitations in their ability to solve relatively simple visual cognitive tasks in comparison to humans (see Ricci et al. (2021) for review). Given the vast superiority of animals over state-of-the-art AI systems, it makes sense to turn to brain sciences to find inspiration to leverage brain-like mechanisms to improve the ability of modern deep neural networks to solve complex visual reasoning tasks. Indeed, a recent human EEG study has shown that attention and memory processes are needed to solve same-different visual reasoning tasks (Alamia et al., 2021). This interplay between attention and memory is previously discussed in Buehner et al. (2006); Fougnie (2008); Cochrane et al. (2019) emphasizing that a model must learn to perform attention over the memory for reasoning.

It is thus not surprising that deep neural networks which lack attention and/or memory system fail to robustly solve visual reasoning problems that involve such same-different judgments (Kim et al., 2018). Recent computer vision works (Messina et al., 2021a; Vaishnav et al., 2022) have provided further computational evidence for the benefits of attention mechanisms in solving a variety of visual reasoning tasks. Interestingly, in both aforementioned studies, a Transformer module was used to implement a form of attention known as self-attention (Cheng et al., 2016; Parikh et al., 2016). In such a static module, attention mechanisms are deployed in parallel across an entire visual scene. By contrast, modern cognitive theories of active vision postulate that the visual system explores the environment dynamically via sequences of attention shifts to select and route task-relevant information to memory. Psychophysics experiments (Hayhoe, 2000) on overt visual attention have shown that eye movement patterns are driven according to task-dependent routines.

Inspired by active vision theory, we describe a dynamic attention mechanism, which we call *guided attention*. Our proposed Guided Attention Module for (visual) Reasoning (GAMR) learns to shift attention dynamically, in a task-dependent manner, based on queries internally generated by an LSTM executive controller. Through extensive experiments on the two visual reasoning challenges, the Synthetic Visual Reasoning Test (SVRT) by Fleuret et al. (2011) and the Abstract Reasoning Task (ART) by Webb et al. (2021), we demonstrate that our neural architecture is capable of learning complex compositions of relational rules in a data-efficient manner and performs better than other state-of-the-art neural architectures for visual reasoning. Using explainability methods, we further characterize the visual strategies leveraged by the model in order to solve representative reasoning tasks. We demonstrate that our model is compositional – in that it is able to generalize to novel tasks efficiently and learn novel visual routines by re-composing previously learned elementary operations. It also exhibit zero shot generalization ability by translating knowledge across the tasks sharing similar abstract rules without the need of re-training.

**Contributions** Our contributions are as follows:

- We present a novel end-to-end trainable guided-attention module to learn to solve visual reasoning challenges in a data-efficient manner.

- We show that our guided-attention module learns to shift attention to task-relevant locations and gate relevant visual elements into a memory bank;

- We show that our architecture demonstrate zero-shot generalization ability and learns compositionally. GAMR is capable of learning efficiently by re-arranging previously-learned elementary operations stored within a reasoning module.

- Our architecture sets new benchmarks on two visual reasoning challenges, SVRT (Fleuret et al., 2011) and ART (Webb et al., 2021).

## 2 PROPOSED APPROACH

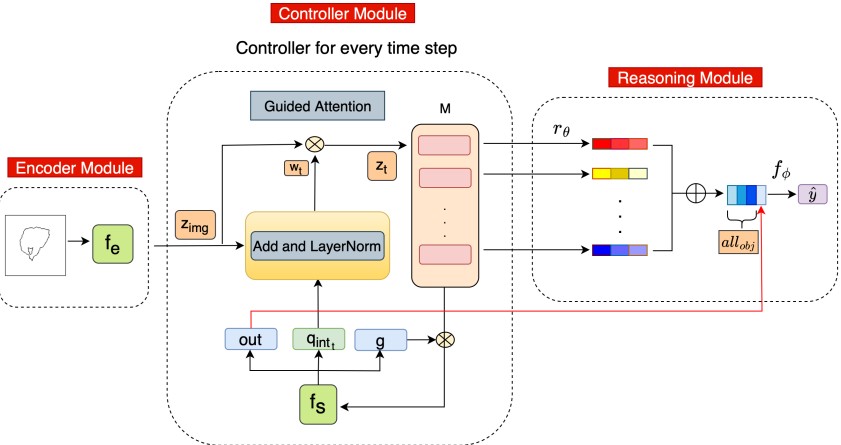

Figure 1: Our proposed *GAMR* architecture is composed of three components: an *encoder* module ($f_e$) builds a representation ($z_{img}$) of an image, a *controller* guides the attention module to dynamically shift attention, and selectively routes task-relevant object representations ($z_t$) to be stored in a memory bank ($M$). The recurrent controller ($f_s$) generates a query vector ($q_{int_t}$) at each time step to guide the next shift of attention based on the current fixation. After a few shifts of attention, a *reasoning* module ($r_\theta$) learns to identify the relationships between objects stored in memory.

Our model can be divided into three components: an encoder, a controller, and a relational module (see Figure 1 for an overview). In the **encoder module**, a low dimensional representation ($z_{img}$) for an input image ($x_{in}$) is created. It includes a feature extraction block ($f_e$) which is composed of five convolutional blocks (SI Figure S1). The output of the module is denoted as $z_{img} \in \mathcal{R}^{(128,hw)}$ (with $h$ height and $w$ width). We applied instance normalization (*iNorm*) (Ulyanov et al., 2016) over $z_{img}$

before passing it to the controller for further processing without which the network even struggles to learn even simple relations.

**Controller module** is composed of two blocks: a recurrent neural network ($f_s$) which generates an internal query to guide the attention spotlight over the task relevant features. These features are used to generate context vector ($z_t$) with the help of the second block, i.e., guided attention (*GA*) and are sent to the memory bank (*M*).

After $z_{img}$ is built, a *guided-attention* block is used to extracts the relevant visual information from the input image in a top-down manner at each time step ($t$). This block generates the context vector ($z_t$) to be stored in the memory bank (*M*) along with all the previous context vectors. They are subsequently accessed again later by a reasoning module. This memory bank is inspired by the differential memory used in Webb et al. (2021).

---

**Algorithm 1** Guided Attention Model for (visual) Reasoning (*GAMR*). $LN$ represents layer normalization (Ba et al., 2016) ($||$) indicates the concatenation of two vectors, forming a new vector. $\{,\}$ indicates the concatenation of a matrix and a vector, forming a matrix with one additional row. $\odot$ represents element-wise multiplication and $\{.\ \}$ represents the product between a scalar and a vector. (h,w) corresponds to the height and width of the feature map obtained from the encoder ($f_e$).

---

$k_{r_{t=1}} \leftarrow 0$ $\qquad\qquad\qquad\qquad\qquad\qquad \triangleright \in \mathcal{R}^{128}$
$h_{t=1} \leftarrow 0$ $\qquad\qquad\qquad\qquad\qquad\qquad\ \triangleright \in \mathcal{R}^{512}$
$M_{t=1} \leftarrow \{\}$
$z_{img} \leftarrow f_e(x_{in})$ $\qquad\qquad\qquad\qquad \triangleright \in \mathcal{R}^{(hw,128)}$
**for** t in 1...T **do**
$\quad out,\ g,\ q_{int_t},\ h_t \leftarrow f_s(h_{t-1},\ k_{r_{t-1}})$ $\qquad \triangleright$
$\quad out \in \mathcal{R}^{512}, g \in \mathcal{R}^{128}, q_{int_t} \in \mathcal{R}^{128}$
$\quad w_t \leftarrow LN(z_{img} + q_{int_t}.repeat(hw, axis = 1)) \triangleright \in \mathcal{R}^{(hw,128)}$
$\quad z_t \leftarrow (z_{img} \odot w_t.sum(axis = 1)).sum(axis = 1) \triangleright \in \mathcal{R}^{128}$
$\quad$**if** t is 1 **then**
$\quad\quad k_{r_t} \leftarrow 0$
$\quad$**else**
$\quad\quad k_{r_t} \leftarrow g \odot M_{t-1}.sum(axis = 1)$
$\quad$**end if**
$\quad M_t \leftarrow \{M_{t-1},\ z_t\}$ $\qquad\qquad\qquad \triangleright \in \mathcal{R}^{(t,128)}$
**end for**
$all_{obj} \leftarrow r_\theta(\sum_{i,j=1}^{T}(M_{v_i},\ M_{v_j}))$
$\hat{y} \leftarrow f_\phi(all_{obj}\ ||\ out)$

---

In the guided attention block, an attention vector ($w_t \in \mathcal{R}^{128}$) is obtained by normalizing the addition of encoded feature ($z_{img}$) with internally generated query ($q_{int_t}$) fetched by $f_s$. This normalized attention vector is used to re-weight the features at every spatial location of $z_{img}$ to generate the context vector $z_t \in \mathcal{R}^{128}$. The recurrent controller ($f_s$) uses a Long Short-Term Memory (LSTM) to provide a query vector ($q_{int_t} \in \mathcal{R}^{128}$) in response to a task-specific goal in order to guide attention for the current time step $t$. $f_s$ also independently generates a gate vector ($g \in \mathcal{R}^{128}$) and output vector ($out \in \mathcal{R}^{512}$) with the help of linear layers. The gate ($g$) is later used to shift attention to the next task-relevant feature based on the features previously stored in $M$. On the other hand, the decision layer uses the output vector ($out$) to produce the system classification output (SI Figure S3).

The **relational module** is where the reasoning takes place over the context vector ($z_t$) stored in the memory bank ($M$). This module is composed of a two layered MLP ($r_\theta$) which produces a relational vector ($all_{obj}$) similar to the relational network (Santoro et al., 2017). As we will show in section 5, $r_\theta$ learns elementary operations associated with basic relational judgments between context vectors ($z_t$) stored in the memory ($M$). It is concatenated with the output ($out$) of the controller ($f_s$) at the last time step ($t = T$) and passed through the decision layer ($f_\phi$) to predict the output ($\hat{y}$) for a particular task. We have summarized the steps in Algorithm 1.

## 3 METHOD

**Dataset** The SVRT dataset is composed of *twenty-three* different binary classification challenges, each representing either a single rule or a composition of multiple rules. A complete list of tasks with sample images from each category is shown in SI (Figure S17, S18). We formed four different datasets with 0.5k, 1k, 5k, and 10k training samples to train our model. We used unique sets of 4k and 40k samples for validation and test purposes. Classes are balanced for all the analyses.

We trained the model from scratch for a maximum of 100 epochs with an early stopping criterion of 99% accuracy on the validation set as followed in Vaishnav et al. (2022) using Adam (Kingma & Ba, 2014) optimizer and a binary cross-entropy loss. We used a hyperparameter optimization framework *Optuna* (Akiba et al., 2019) to get the best learning rates and weight decays for these tasks and reports the test accuracy for the models that gave the best validation scores.

**Baselines**   For the baselines in this dataset, we compared our architecture performance to a Relational Network ($RN$), a popular architecture for reasoning in VQA. The $RN$ uses the same CNN backbone as *GAMR* with feature maps of dimension $\mathcal{R}^{128,hw}$ where $h = 8$ and $w = 8$. We consider each spatial location of the encoded feature representation as an object (i.e., $N = 8 \times 8 = 64$ object representations). We computed all pairwise combinations between all 64 representations using a shared MLP between all the possible pairs (totalling 4096 pairs). These combinations are then averaged and processed through another MLP to compute a relational feature vector ($all_{obj}$) before the final prediction layer ($f_\phi$). In a sense, *GAMR* is a special case of an $RN$ network endowed with the ability to attend to a task-relevant subset ($N = 4$) of these representations with the help of a controller instead of exhaustively computing all 4,096 possible relations – thus reducing the computing and memory requirements of the architecture very significantly.

As an additional baseline model, we used 50 layered *ResNet* (He et al., 2016) and its Transformer-based self-attention network (*Attn-ResNet*) introduced in Vaishnav et al. (2022) and follow the training mechanism as defined in the paper. These have been previously evaluated on SVRT tasks (Funke et al., 2021; Vaishnav et al., 2022; Messina et al., 2021b;a). *Attn-ResNet* serves as a powerful baseline because of more free parameters and a self-attention module to compare the proposed active attention component of *GAMR*. In our proposed method, the controller shifts attention head sequentially to individual task-relevant features against a standard self-attention module – where all task-relevant features are attended to simultaneously. We also evaluated memory based architecture, ESBN (Webb et al., 2021) in which we used a similar encoder to that of *GAMR* and pass the images in sequential order with each shape as a single stimulus and the number of time steps as the number of shapes present in the SVRT task. In order to train these models we used images of dimension $128 \times 128$ for architectures such as *RN, ESBN, GAMR* and $256 \times 256$ for *ResNet, Attn-ResNet* (to be consistent with configuration in Vaishnav et al. (2022)). ResNet-50 ($ResNet$) has 23M parameters, Relation Network ($RN$) has 5.1M parameters, ResNet-50 with attention (*Attn-ResNet*) has 24M parameters and *GAMR & ESBN* both have 6.6M parameters.

## 4    BENCHMARKING THE SYSTEM

All twenty-three tasks in the SVRT dataset can be broadly divided into two categories: same-different (SD) and spatial relations (SR), based on the relations involved in the tasks. Same-different tasks (*1, 5, 6, 7, 13, 16, 17, 19, 20, 21, 22*) have been found to be harder for neural networks (Ellis et al., 2015; Kim et al., 2018; Stabinger et al., 2016; 2021; Puebla & Bowers, 2021; Messina et al., 2021a; Vaishnav et al., 2022) compared to spatial relations tasks (*2, 3, 4, 8, 9, 10, 11, 12, 14, 15, 18, 23*).

We analyzed an array of architectures and found that, on average, *GAMR* achieves at least 15% better test accuracy score on $SD$ tasks for 500 samples. In contrast, for $SR$ tasks, the average accuracy has already reached perfection. We find a similar trend for other architectures when trained with different dataset sizes. Overall, RN (GAMR minus attention) and ESBN struggled to solve SVRT tasks even with 10k training samples, pointing towards the lack of an essential component, such as attention. On the other hand, Attn-ResNet architecture demonstrated the second-best performance, which shows its importance in visual reasoning. Results are summarized in Figure 2.

## 5    LEARNING COMPOSITIONALITY

Below, we provide evidence that *GAMR* is capable of harnessing compositionality. We looked for triplets of tasks $(x, y, z)$ such that $z$ would be a composition of tasks $x$ and $y$. We systematically looked for all such available triplets in the SVRT dataset and found three triplets: (*15, 1, 10*), (*18, 16, 10*) and (*21, 19, 25*). All these tasks and their associated triplets are described in SI section S2.5. We study the ability of the network to learn to compose a new relation with very few training samples, given that it had previously learned the individual rules. We first trained the model with tasks $x$ and $y$

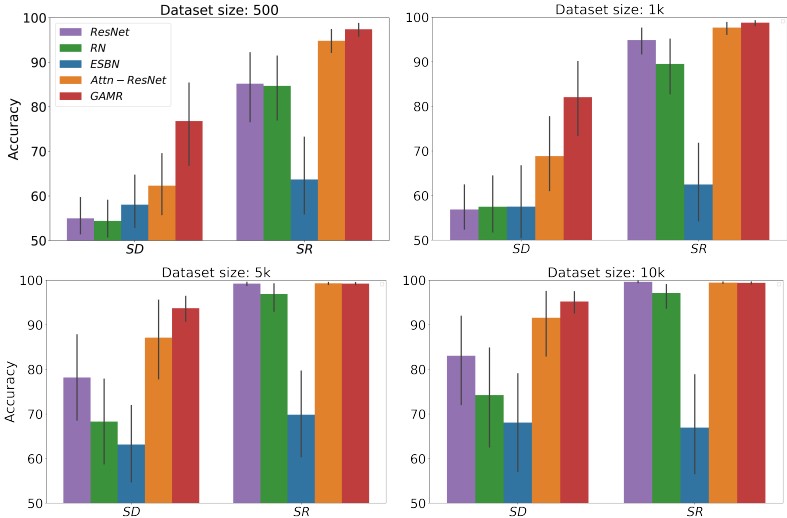

Figure 2: Bar plot analysis for the SVRT tasks grouped in same-different ($SD$) and spatially related ($SR$) tasks. We compared the accuracies of five baseline architectures with *GAMR*. We trained these models with .5k, 1k, 5k and 10k samples.

so that the rules are learned with the help of the reasoning module $r_\theta$ which is a two-layered MLP. We expect that the first layer learns the elementary operation over the context vectors stored in the memory block ($M$), while the second layer learns to combine those operations for the tasks $z$. We freeze the model after training with tasks $x$, $y$ and only fine-tune: (i) a layer to learn to combine elementary operations and (ii) a decision layer ($f_\phi$) on tasks $z$ with ten samples per category and 100 epochs in total. Results are shown in Figure 3 (left). As our baseline, we trained the model from scratch on task $z$ from a triplet of tasks ($x,y,z$) to show that the model is exploring indeed compositionality. We also ran an additional control experiment for compositionality choosing the random pair of tasks ($x=5$, $y=17$) such that the rules are not included in tasks (z) *15, 18*, and *21*. When we evaluated the network in this setup, we found the performance of the network to be at the chance level – aligning with the claim.

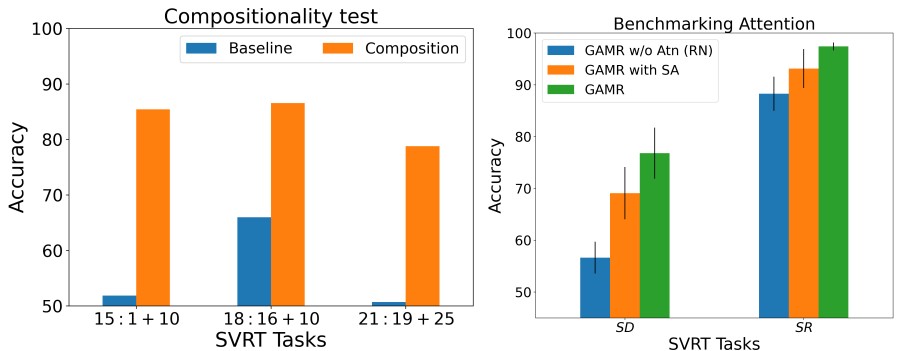

Figure 3: **Compositionality test** (left): We train the model with tasks containing specific rules (e.g., task *1* representing same-different discrimination and task *10* involving identification if the four shapes form a square or not). We show that with its ability to compose already learned rules, *GAMR* can quickly learn with 10 samples per class to adapt to a novel scenario (e.g., task *15* where the rule is to identify if the four shapes forming a square are identical or not). **Benchmarking Guided Attention** (right): We compared the average accuracy over two sub-clusters of SVRT obtained by *GAMR* with its variant when we replaced the guided-attention module with the self-attention (*GAMR-SA*) and when we completely gave away attention and made it a relational reasoning architecture (*GAMR w/o Atn (RN)*).

## 6 ZERO-SHOT GENERALIZATION

We hypothesize that if a model has learned the abstract rule underlying a given task, it should be able to re-use its knowledge of this task on other novel tasks which share a similar rule. To verify that *GAMR* is indeed able to generalize across tasks that share similar rules, we searched for pairs of tasks in SVRT which were composed of at least one common elementary relation (Vaishnav et al., 2022) between them. For example, in pair (*1, 22*), task *1* involves the identification of *two* similar shapes in category 1 and task *22* involves the identification of *three* similar shapes in category 1. In the selected pair, the category that

| Training Task | Test Task | Test Accuracy | | |
|---|---|---|---|---|
| | | GAMR | Attn-ResNet | ResNet |
| 1 | 5 | 72.07 | 53.03 | **73.04** |
| | 15 | **92.53** | 92.07 | 78.87 |
| | 22 | **84.91** | 80.10 | 67.15 |
| 5 | 1 | **92.64** | 85.73 | 92.28 |
| | 15 | **84.36** | 62.69 | 49.95 |
| | 22 | **76.47** | 55.69 | 50.19 |
| 7 | 22 | **83.80** | 79.11 | 50.37 |
| 21 | 15 | **90.53** | 50.00 | 49.76 |
| 23 | 8 | **85.84** | 58.90 | 59.25 |

Table 1: Test accuracy to show if the model learns the correct rules when we train it with a task and test on a different set of SVRT tasks with *GAMR*, Attention with ResNet-50 (Attn-ResNet) and ResNet-50 (ResNet).

judges the similar rule should belong to the same class (let us say category 1 in the above example) so that we test for the right learnability.

We systematically identified a set $x$ of tasks *1, 5, 7, 21, 23* representing elementary relations such as identifying same-different (*1, 5*), grouping (*7*), learning transformation like scaling and rotation (*21*) and learning insideness (*23*). Then we paired them with other tasks sharing similar relations. These pairs are task *1* with each of *5, 15 and 22*, task *5* with each of *1, 15 and 22*. Similarly other pairs of tasks are (*7, 22*), (*21, 15*) and (*23, 8*). We have explained how all these pairs of tasks form a group in SI section S2.3. We separately trained the model on the set $x$ and tested the same model on their respective pairs without fine-tuning further with any samples from the test set (zero-shot classification). We observed that *GAMR* could easily generalize from one task to another without re-training. On the contrary, a chance level performance by *ResNet-50* shows the network's shortcut learning and rote memorization of task-dependent features. In comparison, *GAMR* exhibits far greater abstraction abilities – demonstrating an ability to comprehend rules in unseen tasks without any training at all. We further explored the strategies learned by *GAMR* using attribution methods for all the tasks (see SI section S5). These attribution methods confirm that *GAMR* does indeed use a similar visual routine between the original task for which it was trained and the new task for which it was never trained. Table 1 summarizes these results.

## 7 ABLATION STUDY

**Benchmarking guided attention** We evaluated our guided-attention module (*GAMR*) and compared it with alternative systems with comparable base-architecture but endowed with self-attention (*With-SA*) or no attention and/or memory (*GAMR w/o Atn (RN)*) over 23 SVRT tasks for the same number of time steps. In *GAMR with-SA*, we add a self-attention layer in the guided attention module and all three input vectors to the attention module are the same ($z_{img}$). Our intuition is that at each time step, the self-attention mechanism should provide the mechanism to learn to attend to different objects in a scene. As a side note, *GAMR with-SA* turns out to be similar to ARNe (Hahne et al., 2019) used for solving Raven's tasks. We found that, on average, our Guided Attention Model's relative performance is 11.1% better than its SA counterpart and 35.6% than a comparable system lacking attention (or memory) for $SD$ tasks; similarly, relative improvements for $SR$ tasks are 4.5% and 10.4% higher. It shows that *GAMR* is efficient as it yields a higher performance for the same number (1k) of training samples. Results are shown in Figure 3 (right).

*GAMR* is a complex model with several component, so we now proceed to study what role different components of the proposed architecture play in it's ability to learn reasoning tasks. We studied the effect of these components on SD and SR categories. Our lesioning study revealed that *iNorm* plays a vital role in the model's reasoning and generalization capability even for learning simple rules of $SR$ tasks. Normalizing every sample individually helps the model learn the abstract rules involved in task. We also found that for $SD$ tasks, excluding vector *out* from the decision-making process is detrimental. The t-SNE plot shows that it encodes the independent abstract representation for the SVRT tasks (SI Figure S3). We systematically ran an ablation study to show that each of

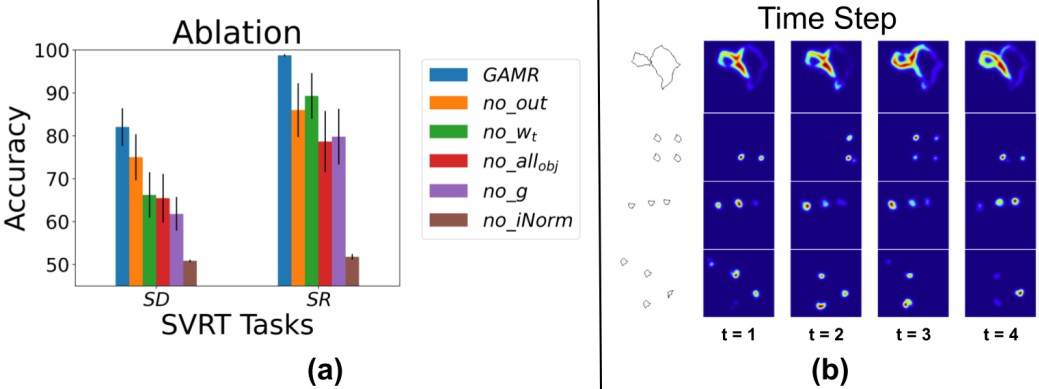

Figure 4: **Ablation studies (a)**: We pruned separate parts of the model, one at a time: controller output ($out$), attention vector ($w_t$), relational vector ($all_{obj}$), feature channel gain factor ($g$) and instance normalization ($iNorm$) and the bar plot show the variation in performance on SD and SR tasks when trained with 1k samples. **Time steps visualization (b)**: Figure showing shift of attention with each time step in a task-dependent manner. In the first row, the task is to answer if the two shapes are touching each other from the outside. At each time step, the network explores the area where the shapes are touching each other. In other rows, attribution maps show the shifts over different shapes in an image. The controller module for the task in respective rows shifts attention across different shapes at each time step.

these components are essential to make it an efficient model. In $no\_all_{obj}$, the model takes a decision based on the final outcome ($out$) of the recurrent module ($f_s$); for $no\_w_t$, output of the attention block is used to obtain $z_t$ instead after projecting it on $z_{img}$; for $no\_g$, equal weighing is applied to the feature space of the context vectors stored in the memory block. We have summarized the results in Figure 4 (left) and added additional analyses in SI section S2. We also plot the attribution maps of the model in Figure 4 (right) at each time step and show the way in which the model attends to task-dependent features while learning the rule.

## 8 ADDITIONAL EXPERIMENT

**Dataset** Webb et al. (2021) proposed four visual reasoning tasks (Figure S20), that we will hence-forth refer to as the *Abstract Reasoning Task* (ART): (1) a same-different (*SD*) discrimination task, (2) a relation match to sample task (*RMTS*), (3) a distribution of three tasks (*Dist3*) and (4) an identity rule task (*ID*). These four tasks utilize shapes from a set of 100 unique Unicode character images [1]. They are divided into training and test sets into four generalization regimes using different holdout character sets (m = 0, 50, 85, and 95) from 100 characters. We have described training and test samples and different hyper-parameters for all four tasks in SI section S7.

**Baseline models** As a baseline, we chose the ESBN (Webb et al., 2021) along with the two other prevalent reasoning architectures, the Transformer (Vaswani et al., 2017) and Relation Network (RN) (Santoro et al., 2017). These three share a similar encoder backbone as in *GAMR*. We also ran an additional baseline with *ResNet-50* to verify if in a multi-object scenario *GAMR* exploring some biases (like visual entropy) which is otherwise not present when segmented images are passed. In order to make our baselines stronger, we evaluated these models in their natural order, i.e., by passing a single image at a time. We added a random translation (jittering) for the shapes in the area of $\pm 5$ pixels around the center to prevent these architectures from performing template matching. This jittering increases the complexity of the tasks and hence we have to increase the number of time steps from 4 to 6. This caused the differences in results when compared to the implementation in Webb et al. (2021). For *GAMR* and *ResNet-50*, we present task-relevant images together as a single stimulus (SI, Figure S21) while jittering each shape. We have also added ART results where each image is centered and put together in a single stimulus in SI section S8. In order to make our architecture

---

[1] https://github.com/taylorwwebb/emergent_symbols

choose one option from multiple stimuli (*RMTS*: 2, *Dist3* and *ID*: 4), we concatenate the relational vector ($all_{obj}$) for every stimulus and pass them to a linear layer for final decision.

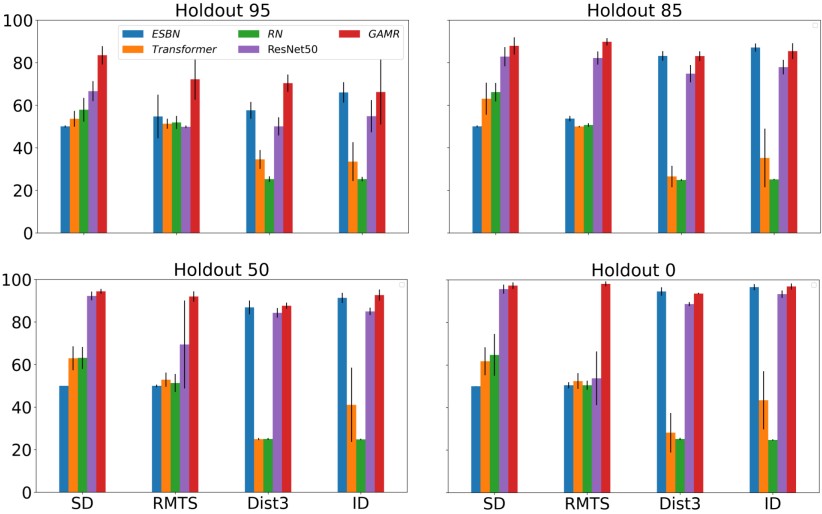

Figure 5: **ART:** Average performance of *GAMR* and other baselines over 10 runs for different holdout values (m = 0, 50, 85, 95). These models are evaluated on four tasks, i.e., Same-Different (SD), Relation match to sample (RMTS), Distribution of 3 (Dist3) and Identity rules (ID).

**Results** We found a near-chance level (50%) accuracy for all the baseline models and in all the four generalization regimes for the *SD* and *RMTS* tasks (Figure 5) which otherwise performed with perfection when images were centered and passed through the same models. However, our proposed architecture is robust to handle this jittering, as shown in SI Figure S22 where we compare its performance when images are not jittered. For the other two tasks, *Dist3* and *ID*, baseline models performed better than the chance level (25%). *ESBN* showed an increasing trend in accuracy for progressively easier generalization conditions approaching 0 holdouts. This points toward the fact that the first three shapes in both tasks allow *ESBN* to consider a translation factor while comparing the next three shapes, letting it choose the correct option appropriately. *RN* and *Transformer* consistently struggled to generalize. *ESBN* (memory-based model) performance on SD tasks in both the visual reasoning datasets show that attention is needed for reasoning.

## 9 RELATED WORK

Multiple datasets have been used to assess the visual reasoning ability of neural networks. One of the first challenges included the SVRT. Recently introduced Raven style Progressive Matrix datasets, RAVEN (Zhang et al., 2019), PGM Barrett et al. (2018), focuses on learning nearly seven unique rules and choose one of the eight choices. However, it was found that the dataset was seriously flawed as it was later found that neural architectures could solve tasks by leveraging shortcuts (Hu et al., 2020; Spratley et al., 2020) which were later removed in I-RAVEN (Hu et al., 2021). Prior work (Kim et al., 2018; Vaishnav et al., 2022; Messina et al., 2021a) on SVRT studies has focused on the role of attention in solving some of these more challenging tasks. In SVRT, tasks that involve same-different (SD) judgements appear to be significantly harder for neural networks to learn compared to those involving spatial relation judgement (SR) (Stabinger et al., 2016; Yihe et al., 2019; Kim et al., 2018) (see Ricci et al. (2021) for review). Motivated by neuroscience principles, Vaishnav et al. (2022) studied how the addition of feature-based and spatial attention mechanisms differentially affects the learnability of the tasks. These authors found that SVRT tasks could be further taxonomized according to their differential demands for these two types of attention. In another attempt to leverage a Transformer architecture to incorporate attention mechanisms for visual reasoning, Messina et al. (2021a) proposed a recurrent extension of the classic Vision Transformer block (R-ViT). Spatial

attention and feedback connections helped the Transformer to learn visual relations better. The authors compared the accuracy of four same-different (SVRT) tasks (tasks *1,5,20,21*) to demonstrate the efficacy of their model.

With the introduction of Transformer architecture, attention mechanisms started gaining popularity in computer vision. They can either complement (Bello et al., 2019; Vaishnav et al., 2022; d'Ascoli et al., 2021) or completely replace existing CNN architectures (Ramachandran et al., 2019; Touvron et al., 2021; Dosovitskiy et al., 2020). Augmenting the attention networks with the convolution architectures helps them explore the best of the both worlds and train relatively faster. In contrast, stand-alone attention architecture takes time to develop similar inductive biases similar to CNN. As initially introduced by Vaswani et al. (2017), Transformer uses a self-attention layer, followed by residual connection and layer normalization and a linear projection layer to compute the association between input tokens. We used a similar system (SI Figure S2) where instead of using a self-attention module, in GAMR, feature-based attention vector (an internally generated query) is obtained via an LSTM to guide the attention module to the location essential for the task and we thus call the model as *guided-attention*. Since there could be more than one location where the model will attend, we then implemented a memory bank. A more closely aligned model with the human visual system is proposed by Mnih et al. (2014) – Recurrent Attention Model (RAM). It learns a saccadic policy over visual images and is trained using reinforcement learning to learn policies (see SI S2.1 for discussion and its performance on SVRT task). The Mnih et al system constitutes an example of overt attention. Conversely, GAMR constitutes an example of a covert attention system and assumes a fixed acuity.

We took inspiration for the memory bank from ESBN (Webb et al., 2021), where mechanisms for variable binding and indirection were introduced in architecture for visual reasoning with the help of external memory. Variable binding is the ability to bind two representations, and indirection is the mechanism involved in retrieving one representation to refer to the other. While ESBN was indeed a source of inspiration, we would like to emphasize that GAMR constitutes a substantial improvement over ESBN. First and foremost, ESBN lacks attention. It requires items/objects to be passed serially one by one and hence it cannot solve SVRT or any other multi-object visual reasoning problems. In a sense, the approach taken in ESBN is to assume an idealized frontend that uses hard attention to perfectly parse a scene into individual objects and then serially pipe them through the architecture. This is where our work makes a substantial contribution by developing an attention front-end (which is soft and not hard) to sequentially attend to relevant features and route them into memory. We tested this template-matching behavior of the ESBN architecture by training it in the presence of Gaussian noise and spatial jittering. It led to a chance-level performance (refer to SI section S8 for more details). Here, we build on this work and describe an end-to-end trainable model that learns to individuate task-relevant scenes and store their representations in memory to allow the judging of complex relations between objects. Finally, our relational mechanism is inspired by the work in Santoro et al. (2017) that introduced a plug-and-play model for computing relations between object-like representations in a network.

## 10    Conclusion and limitations

In this paper, we described a novel Guided Attention Module for (visual) Reasoning (*GAMR*) to bridge the gap between the reasoning abilities of humans and machines. Inspired by the cognitive science literature, our module learns to dynamically allocate attention to task-relevant image locations and store relevant information in memory. Our proposed guided-attention mechanism is shown to outperform the self-attention mechanisms commonly used in vision transformers. Our ablation study demonstrated that an interplay between attention and memory was critical to achieving robust abstract visual reasoning. Furthermore, we demonstrated that the resulting systems are capable of solving novel tasks efficiently – by simply rearranging the elemental processing steps to learn the rules without involving any training. We demonstrated GAMR's versatility, robustness, and ability to generalize compositionality through an array of experiments. We achieved state-of-the-art accuracy for the two main visual reasoning challenges in the process. One limitation of the current approach is that it currently only deals with a fixed number of time steps. Training the model with four time steps was sufficient to solve all SVRT tasks efficiently. However, a more flexible approach is needed to allow the model to automatically allocate a number of time steps according to the computational demand of the task. GAMR is also limited to covert attention unlike biological system where both covert and overt attention are demonstrated.

## ACKNOWLEDGMENTS

We want to thank Jonathan D. Cohen (Princeton University), Taylor Webb (UCLA), Minju Jung (Brown University) and Aimen Zerroug (ANITI, Brown University) and anonymous reviewers for helpful discussions. We also want to acknowledge Thomas Fel's (ANITI, Brown University) help in feature visualization.

This work is funded by NSF (IIS-1912280) and ONR (N00014-19-1-2029) to TS. Additional support was provided by the ANR-3IA Artificial and Natural Intelligence Toulouse Institute (ANR-19-PI3A-0004). Computing resources used supported by the Center for Computation and Visualization (NIH S10OD025181) at Brown and CALMIP supercomputing center (Grant 2016-p20019, 2016-p22041) at Federal University of Toulouse Midi-Pyrénées.

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

# Supplementary Information

## S1    IMPLEMENTATION DETAILS

**Convolutional Backbone**    We used the same encoder ($f_e$) as a backbone for the following architectures: *GAMR, ESBN* and *RN*. This encoder is built with the help of six convolutional blocks each as described in Table S1.

Table S1: **Convolutional backbone architecture**. in_ch and out_ch represent the number of input and output channels for a convolutional layer. $\times$ signifies the number of times the above-mentioned block consisting of [convolution–BatchNorm–ReLU] is repeated.

| Layer Name | Output size | Configuration |
|---|---|---|
| Conv_0 | 128 x 128 | in_ch = 3, out_ch = 64, kernel_size = 3, stride = 1, padding =1
Batchnorm(64)
ReLU
$\times$ 1

in_ch = 64, out_ch = 64, kernel_size = 3, stride = 1, padding =1
Batchnorm(64)
ReLU
$\times$ 1 |
|  | 64 x 64 | MaxPooling |
| Conv_1 | 64 x 64 | in_ch = 64, out_ch = 64, kernel_size = 3, stride = 1, padding =1
Batchnorm(64)
ReLU
$\times$ 4 |
|  | 32 x 32 | MaxPooling |
| Conv_2 | 32 x 32 | in_ch = 64, out_ch = 128, kernel_size = 3, stride = 1, padding =1
Batchnorm(128)
ReLU
$\times$ 1

in_ch = 128, out_ch = 128, kernel_size = 3, stride = 1, padding =1
Batchnorm(128)
ReLU
$\times$ 3 |
|  | 16 x 16 | MaxPooling |
| Conv_3 | 16 x 16 | in_ch = 128, out_ch = 128, kernel_size = 3, stride = 1, padding =1
Batchnorm(128)
ReLU
$\times$ 4 |
|  | 8x8 | MaxPooling |
| Conv_4 | 8x8 | in_ch = 128, out_ch = 128, kernel_size = 3, stride = 1, padding =1
Batchnorm(128)
ReLU
$\times$ 3

in_ch = 128, out_ch = 256, kernel_size = 3, stride = 1, padding =1
Batchnorm(256)
ReLU
$\times 1$ |
| Conv_5 | 8x8 | in_ch = 256, out_ch = 256, kernel_size = 3, stride = 1, padding =1
Batchnorm(256)
ReLU
$\times 4$ |
| Conv Layer | 8x8 | in_ch = 256, out_ch = 128, kernel_size = 1, stride = 1, padding =0 |

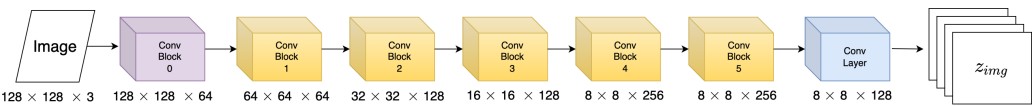

Figure S1: Encoder module ($f_e$) of *GAMR*

**GAMR**    The architecture is described in Algorithm 1. An encoder module ($f_e$) is designed as shown in Table S1. The LSTM in the controller layer ($f_s$) is a single layered with 512 units. The output ($out \in \mathcal{R}^{512}$) of the controller is used to obtain the gate vector ($g$) with the help of a linear layer and ReLU non-linearity. An internal query ($q_{int_t}$) is also generated with the help of a linear layer without any non-linearity.

The relational module $r_\theta$ is a two-layered MLP. Both these layers have ReLU non-linearity. The first hidden layer is of size 512 dimension and the second layer is of size 256 dimension. The output from the second layer is summed after $t$ time steps. They are concatenated with the *out* vector and passed to the decision layer for generating $\hat{y}$. No non-linearity is associated with the output layer.

**Relational Network (RN)**  In this architecture, once the $z_{img} \in \mathcal{R}^{(64,128)}$ is obtained from the encoder block ($f_e$), we directly move to the relational module part ($r_\theta$) as defined above. Instance normalization is also applied to the $z_{img}$ as done in $GAMR$. In the relational module, all the pairwise combinations are obtained for each spatial location of the $z_{img}$ vector making 4096 (64×64) combinations in total. The output from this module is sent to the decision layer for final prediction $\hat{y}$.

**GAMR-SA**  In the self-attention variant of GAMR, instead of passing $q_{int_t}$ in the guided attention module, we pass $z_{img}$ as key, query and value vectors and calculate the self-attention at that particular time step. All the other components of the architecture are kept the same as in $GAMR$.

**GAMR for ART tasks**  ART challenge consists of 4 types of visual reasoning tasks (*SD, RMTS, Dist3, ID*). GAMR assumes a multi-object setup for all these four tasks, i.e., multiple images embedded in a single stimulus. This makes the $SD$ task a binary classification challenge (similar to SVRT challenge task *1*). However, for the other three tasks, we take the relational vector ($r_\theta \in \mathcal{R}^{128}$) and concatenate the respective relational vector for all the input stimuli (RMTS – 2, Dist3/ID – 4). Later, this concatenated vector is used to predict the final outcome with a linear layer. We used a similar technique for ResNet-50 where the linear layer after average pooling is used to generate $\mathcal{R}^{128}$ which is concatenated and passed to a decision layer.

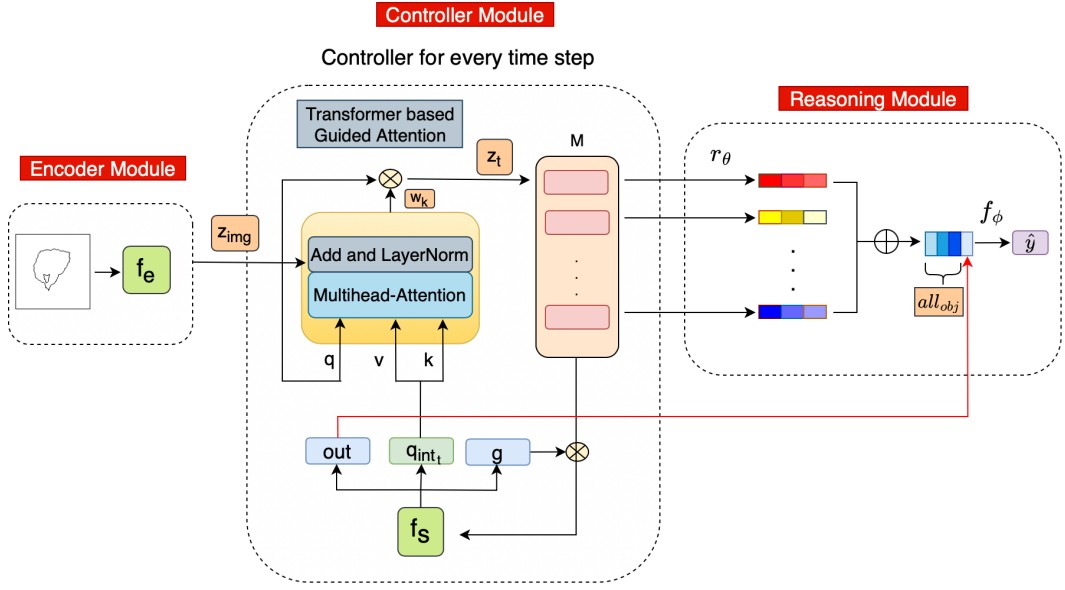

Figure S2: *GAMR approximation with Transformer:* Equivalent representation of GAMR architecture where the guided attention model can be approximated to Transformer encoder layer where the query ($q$), key ($k$) and value ($v$) are passed as $z_{img}$, $q_{int_t}$ and $q_{int_t}$ respectively.

## S2  IN DEPTH ANALYSIS OF *GAMR*

### S2.1  RELATION WITH HUMAN VISUAL REASONING

There is evidence for overt attention shifts (i.e., associated with saccades) and covert (i.e., without saccades). Both are closely related and overt attention is needed to be deployed at a location before

a saccade can be made at that location (see premotor theory of attention (Rizzolatti & Craighero, 2010). From a computational standpoint, the only real difference between overt and covert attention exists for vision systems endowed with a fovea (i.e., a region of greater visual acuity at the center of the retina). Mnih et al. (2014) system constitutes an example of overt attention (and indeed the authors take inputs at a greater resolution at attended locations). Conversely, GAMR constitutes an example of a covert attention system and hence assumes a fixed acuity. Hence, the success of our approach provides evidence for the benefits of attention mechanisms per se and is not confounded by improvements that could arise because of greater acuity. It is also important to note that according to the premotor theory of attention covert attention is required prior to overt shifts/saccades. It should be possible to extend GAMR in a biologically plausible way by explicitly enabling saccades to overtly attended regions and by taking into account different acuities at the fovea vs periphery.

**Comparison with RAM model**    Mnih et al. (2014) proposed a Recurrent Attention Model (RAM) inspired by the human visual system which learns a saccadic policy over visual images. We ran the Mnih et al. model for the easier SVRT tasks. We found that the network is unable to learn task *2* requiring to find whether the smaller one of the two shapes is in the center of the larger one or around the boundary. The results are shown in Table S2. We used an image resolution of 60x60 and train the model for 200 epochs with 500 samples. We also trained the model with increased image resolution up to $128 \times 128$ but this led to a very significant drop in accuracy to near chance level.

In addition, we would like to point out that the architecture proposed by Mnih et al. is non-differentiable and it is trained using reinforcement learning to learn policies, which makes it significantly different from our proposed end-to-end differentiable model. In short, we believe that there exist both qualitative and quantitative differences between the two classes of architectures and while the work by Mnih et al was pioneering it is far less expressive than our proposed GAMR.

Table S2: **Recurrent Attention Model (RAM)**: Test and validation accuracy of RAM over SVRT task *2*

| Accuracy | Glimpse | | | |
|---|---|---|---|---|
| | **4** | **5** | **6** | **10** |
| Best validation accuracy | 78.00 | 72.00 | 74.00 | 72.00 |
| Test Accuracy | 62.26 | 61.55 | 71.21 | 58.98 |

## S2.2 CHARACTERIZING THE LEARNED STRATEGIES WITH EXPLAINABILITY METHODS

We used three attribution methods: *Integrated-gradients* (Sundararajan et al., 2017), *Saliency* (Simonyan et al., 2014) and *SmoothGrad* (Smilkov et al., 2017). Common to these methods is the use of the gradient of the classification units w.r.t. the input image. These methods help characterize the visual features that are diagnostic for the network prediction (e.g., SI section S4).

## S2.3 LEARNING RULES

**Rules involving Same-Different (SD) identification**    In task *1*, the challenge is to differentiate between the two shapes and recognize if these two shapes are the same or different. When we trained the model on task *1*, it (1) learned a mechanism to attend to the two shapes presented in an image; (2) learned to judge the similarity between the two shapes. To test the models' generalization ability, we selected tasks demanding similar judgment to decide. There are three of those from the set of 23 satisfying this criterion, *5, 15* and *22*. In task *15*, four shapes form a square; in category 1, all these shapes are identical. Compositions involved in this task are identifying the same shape and counting them to make a correct decision. *GAMR* has shown its ability to attend to all the identical shapes (Figure S11) and make a correct decision with 92.53% accuracy. The model learns a similar ability when trained with task *21*. It has two shapes that are in rotation, scaled, or translated form. The reason task *21* could solve task *15* is same as before with *1*.

In task *22*, there are three shapes aligned in a row. All of them are identical in category 1. Again, the model must count the number of identical shapes and make the correct decision if the count reaches 3. In this task, our model achieves 84.91% accuracy. Another task taking the decision based on counting is *5*. The model needs to identify if two pairs of identical shapes are present in the image.

However, when we evaluate ResNet-50 on these three tasks, we believe that the model must have found a shortcut to make a decision. ResNet-50 has learned a strategy to count only one pair of identical shapes, and it takes the decision as soon as it finds that identical pair in tasks *5, 15, 22* and performs better than the chance level.

Next, we try to understand the learning of the model trained on task *5*. The model trained with task *5* has learned to identify two pairs of identical shapes from a group of four. We found that the network is exploring a shortcut to solve this task. Rather than learning to find both pairs of identical shapes, it learned to attend to only one pair to take a judgment (Figure S5). Still, the model's routine helps extend this mechanism and attends to more than two shapes if shown in the same stimulus. At the same time, ResNet-50 uses its template matching technique to explore one pair of identical shapes, which helps it to solve task *1* but does not help on tasks *15 and 22*.

In task *7*, the model learns to identify and count identical shapes in an image. There are three pairs of two identical shapes vs. two pairs of three identical shapes, as seen in Figure S5. We tested this model on task *22* and obtained 83.8% accuracy. Here the rule is to identify if three identical shapes form a row or not. However, a model trained on task *22* cannot solve task *7* because it explores a shortcut in solving task *22*. It compares two shapes and arrives at a decision (Figure S6), whereas, in task *7*, grouping and counting are required to make a decision. ResNet-50, on the other hand, might still be looking for only one pair of identical shapes (there are three pairs in total) to make a decision. Thus fail to adapt to a novel task.

**Rules involving Spatial Relations**  In addition to solving SD tasks, we show that *GAMR* also learns the routines when the judgment requires spatial reasoning. In tasks *11* and *2*, the challenge is to inspect if the two shapes are touching/around the corner or not. One of the smaller shapes in these tasks is towards the boundary from inside (task *2*) or touching from outside (task *11*). Training one and testing the other yields > 95% accuracy. In Figure S14, we can see from the attribution maps that the model is focusing on the area where the two shapes are touching to reach a decision. In task *23*, the challenge is to learn insideness. Task *23* has two small and one large shape; both the small shapes are either inside or outside in one category vs. one shape inside and one outside in category 2. We test the model on task *8*, which has two shapes, one inside the other and identical in one category vs. outside in category 2. When we train a model on task *23*, it learns the judgment of insideness and also counts the number of shapes present inside. If we evaluate such a model on task *8*, it quickly figures out instances when the shape is inside or outside (probably not identical).

For all these experiments, we take 10k samples for training the network for tasks *1, 5, 7, 21, 23, 11, 2* and then perform the zero-shot generalization on a novel set of tasks.

## S2.4    REVERSE CURRICULUM

In this part of the experiment, we go beyond making a decision based on the learned rule and analyze *GAMR*'s ability to parse those rules individually and evaluate tasks containing simpler rules. We trained *GAMR* on a complex task and tested it on its elementary subset. For example, in task *19*, the rules involved are to learn the scale invariance between two shapes in a stimulus. After training the model on task *19*, we tested for the case where the scale factor is 1 (not seen during training), i.e., task *1*, and obtained 94.05% accuracy. Similarly, we took another task *21*, where there are two elementary operations involved, scaling and rotation together (non-zero values). To see if *GAMR* has the ability to parse this complex relation we tested the model trained on task *21* on task *19* (scaling = $x$ and rotation = 0) and task *1* (scaling = 1 and rotation = 0) and obtain 86.95% and 90.86% respectively. It shows *GAMR*'s unique ability to parse a complex set of relations into independent simple elementary unseen relations.

## S2.5    LEARNING COMPOSITION

We selected group corresponding to each tasks (*15, 18, 21*) used for composition. Task *15* has four shapes forming a square and are identical. It can be composed of task *1* helping to identify the same shapes and task *10* which helps to learn if the four shapes are forming a square. In task *18*, a rule is needed to be learned related to symmetry along the perpendicular bisector of the image. It can be taken as a composition of task *16* which requires learning mirror reflection of the image along the perpendicular bisector of the image and task *10* where in which symmetry could be discovered

in between 4 shapes (forming a square). At last, we took task *21* which involves both scaling and rotation between two shapes in an image. As its compositional elements, we designed a variant where there is only rotation and no scaling and represented it with *25* and combined it with another counterpart of *21* where there is scaling and no rotation, i.e., task *19*.

## S2.6   ABSTRACT VARIABLE

As seen in our ablation study (section 7), one of the fundamental components responsible for learning complex rules is *out*, and abstaining from it makes the model struggle. We did a t-SNE analysis on this variable using the first 20 principal components from its 512-dimensional vector as these components contribute to 95% of the variance. We analyzed 200 samples from each task of a class-balanced validation dataset. None of these were included in the training. These components encode the abstract variable for each task and have separable boundaries from each other, as seen in Figure S3.

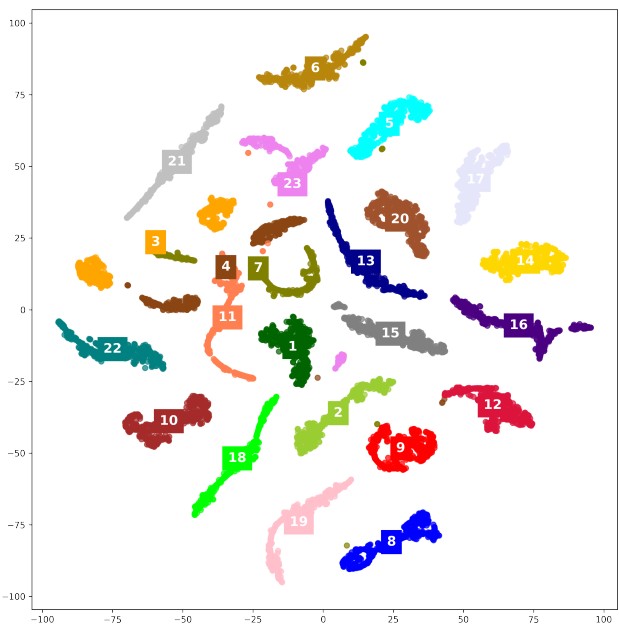

Figure S3: t-SNE plot of the first 20 components of the abstract variable (*out*) obtained via PCA for all 23 SVRT tasks. Those components have a cumulative variance of 95%. Each cluster can be clearly identified from other clusters representing different relations learned. Tasks are represented as labels with the same colored box around them placed in the mean location of the cluster.

## S2.7   ROBUSTNESS

We did a study to inspect the robustness of the model. We evaluated the model on two factors: robustness to scale and robustness to the noise. We found that *GAMR* is able to extract the correct rules in both of these scenarios.

**Scale**   We analyzed the performance of the model with changing scale. We train the model with images of resolution $128 \times 128$ and test them at scale 1.5 times the original, i.e., $192 \times 192$. We found that the model can generalize at this scale to various tasks such as *1, 9, 11, 14, 15, 17*. In Table S2, we compared the performances of the model trained on tasks at scale (s=1) and tested on task with s=1 with tested on task with s=1.5 for two different training sets, 5k and 10k. We did not use any augmentation relating to changing scale during the training, which might push the model to learn such scale-invariant representations.

In Figure S4, we demonstrate an example of task *1* highlighting the region considered for taking decision at s=1.5.

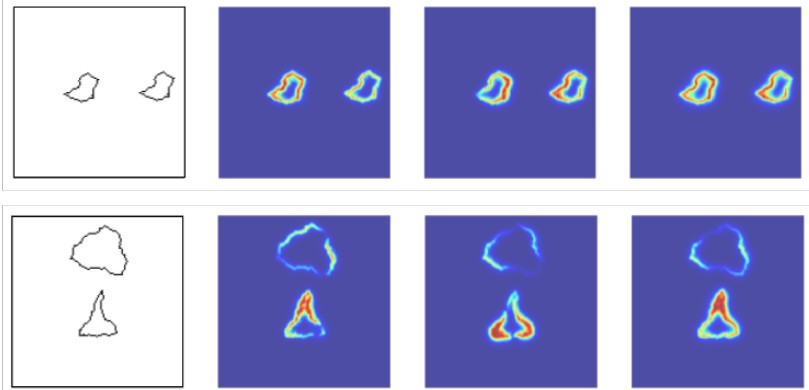

Figure S4: Scale invariance shown by *GAMR* when it is trained with images of dimension $128 \times 128$ and tested on images of dimension $192 \times 192$ for task *1*. Attribution map shows that when two same shapes are present in an image, the model still tries to compare each part of both the shapes vs when two different shapes are present, it focuses its attention on the most salient part i.e. corners. We show maps from three different attribution methods, Integrated Gradients, Saliency and SmoothGrad.

Table S2: Test accuracies when the model is trained on tasks at scale (s) = 1 with 10k and 5k samples and tested on the same tasks at scale (s) = 1.5 without any fine-tuning.

| TRAINING TASKS | TEST ACCURACY | | | |
|---|---|---|---|---|
| | 5K SAMPLES | | 10K SAMPLES | |
| | s=1 | s=1.5 | s=1 | s=1.5 |
| 1 | 94.77 | 71.01 | 97.34 | 75.09 |
| 9 | 99.00 | 70.46 | 99.58 | 72.11 |
| 11 | 99.85 | 90.89 | 99.75 | 91.05 |
| 14 | 98.66 | 71.84 | 99.62 | 73.45 |
| 15 | 98.76 | 73.02 | 98.59 | 68.66 |
| 17 | 93.09 | 72.98 | 95.13 | 74.98 |

**Noise** We even checked the robustness of *GAMR* to noise. We added Gaussian white noise to the dataset to perform this experiment with 0 mean and .05 variance. We found a negligible difference in the test accuracy of the model with noise when compared to the case without any noise (Figure S25, S26). We even went further to change the type of noise to salt and pepper while testing the same model. Nevertheless, the model learned the rules delivering nearly the same test accuracy as before.

## S3 VISUAL ROUTINES

Every instant vast amount of visual information is presented to the sensory systems in the brain, where it extracts the relevant information. This ability to visually detect, recognize, search or obtain descriptions from the scene helps us execute various daily life tasks. Studies have also shown the insensitivity to changes in the visual scenes during eye movement, also called "changing blindness." This phenomenon indicates that a scene is represented with only a small amount of available information. For doing this, some mechanism should exist in order to select the relevant information based on the current circumstances.

Despite the complexities of these tasks, the underlying principles of visual processing in the brain are relatively simple. For example, even if the visual system has the capability to analyze curves and contours, it is impossible to represent all the possible combinations of these just by using feature

detectors (Ullman, 1984). A similar problem is posed in the visual search scenario. Due to the combinatorial explosion of possible features involved in the formation of a scene, the feedforward detection mechanism finds it hard to understand (Tsotsos, 1990). It shows us that mechanisms exist in our visual system other than simple feature detectors that aid in solving such tasks from the versatility of possible scenes.

To implement such task-dependent visual processing, (Ullman, 1984; 1987) proposed visual routines theory. It is one such method to process visual information beyond creating representations, thereby helping in tasks like object recognition, manipulation and abstract reasoning. A primitive set of operations are defined in Ullman (1987) that are applied to obtain relevant information and spatial relations from a scene. Such a composition of operations is called a *routine*.

The visual routine theory is heavily influenced by the theory of vision by Marr (1982). Ullman and Marr focused on extensive analysis and used a functional approach as a starting point. This type of analysis provides a framework that, on the one hand, can be used in the theory of human cognition while, on the other hand, equally applicable in designing computer vision systems. This theory is based on similar principles that visual systems can solve tasks that feed-forward feature detectors could not do. Tasks are solved with the help of some mechanisms applied on top of these representations obtained by feature detectors (known as base representations). Ullman showed that a model could solve complex tasks if a finite set of simple operations helping the model to reason are sequenced properly (forming a routine). Another similar idea was presented in a utilitarian theory of perception (Ramachandran, 1985). This theory mentions that perception is similar to various parts of the human body, which are collections of ad-hoc pieces, working in their own way as well as together.

Another piece of evidence presented by Roelfsema et al. (2000) showed how the elemental operations required for routines are implemented in the brain. When an image is presented, feed-forward processes in the brain lead to the activity pattern distributed across visual areas. Afterward, the elemental operation comes into play as firing rate modulations. These modulations enforce the grouping of neural responses into coherent object representations. Later, Hayhoe (2000) discussed evidence of routine in vision, indicating that only a small part of the information of a scene is represented in a brain at each moment.

A cognitive blackboard theory is postulated in Roelfsema & de Lange (2016), explaining how the activity of neurons in the early visual cortex helps infer the visual world. The writing operations on the blackboard are compared to the feedback signals from higher cortical areas in the early visual cortex. These feedback signals are read by cortical regions and sub-cortical structures in later processing steps, thereby supporting an exchange of intermediate computational steps.

## S3.1 RELATING VISUAL ROUTINE THEORY WITH *GAMR*:

Visual processing by Ullman can be divided into three stages. First stage is the creation of *base representation*. It is a bottom-up process in which base representations are generated from the visual input. Once these representations are constructed, a visual processor is used to apply sequences of elemental operations to them in the second stage. It extracts the desired information from the base representations in a top-down manner. In the third stage, this extracted information is used to reason in any given task. In humans, the bottom-up process corresponds to the retinotopic maps to capture properties like edge, color, speed and direction of motion. These base representations depend on a fixed set of operations uniformly applied over the entire input without any object/task-specific knowledge.

Such sequences of operations to extract information from the base representations are known as *visual routines*. These elemental operations are task-specific that are used to create *incremental representations* containing information to be used in the next operation. Thus, by applying these routines, incremental representations are constructed using sequences of elemental operations, which are finally used to provide a solution particular to the task. Some examples of these elemental operations are as follows: shifting the processing focus, indexing, marking an object or location for future reference, tracing boundaries, and spreading activation over an area delimited by boundaries. In the shift of processing focus, the spotlight of selective attention helps to shift the processing focus on the image from one part to the other. Indexing enlists all locations where processing focus can shift. Routines are generally generated in response to some internally generated queries by assembly mechanisms in response to some specific goals.

Aligned with this theory, our proposed model has different building blocks. These are base representations ($z_{img}$), incremental representations ($z_t$), memory block to save these incremental representations ($M$), guided attention, and finally reasoning module ($r_\theta$). In our model, *base representation* ($z_{img}$) is created using a convolutional encoder ($f_e$) which is a purely bottom-up process. This representation is fixed and calculated using a feed-forward network for any particular input. $z_{img}$ is used to create an incremental representation ($z_t$) using a visual routine processor represented by our guided attention module. At each time step, the guided attention module takes the input as the base representation ($z_{img}$) and query vector coming from a recurrent module ($f_s$). This guided attention module gives the desired task-relevant feature vector to create $z_t$, which is stored in the memory bank (M) and later used for reasoning.

## S4  ATTRIBUTION MAP VISUALIZATION

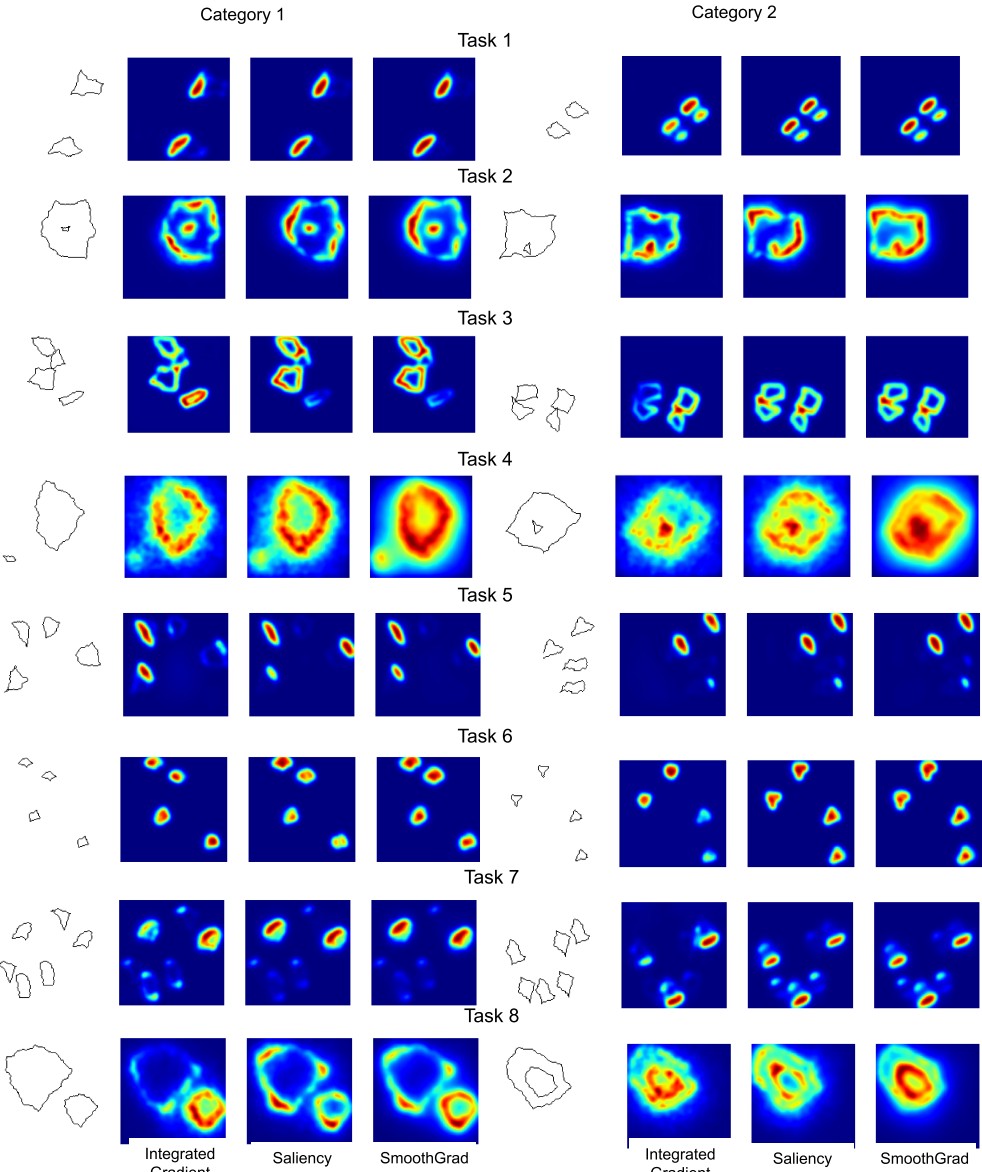

Figure S5: Attribution map visualization for SVRT tasks. The map represents the area of the image, which are significant in decision-making for *GAMR*. Tasks represented are from *1* to *8*. We used three different attribution methods here Integrated Gradients, Saliency and SmoothGrad.

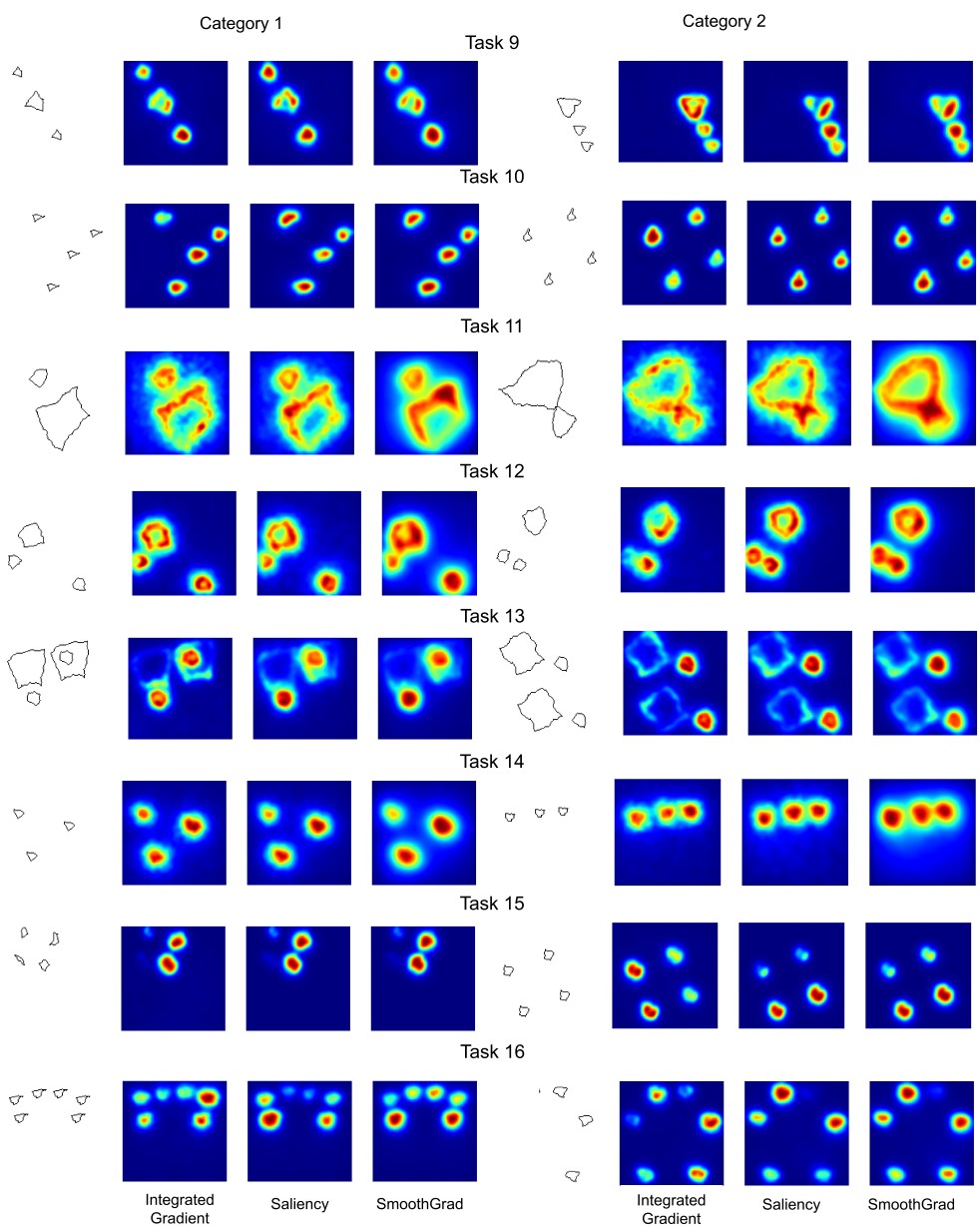

Figure S6: Attribution map visualization for SVRT tasks. The map represents the area of the image, which are significant in decision-making for *GAMR*. Tasks represented are from *9* to *16*. We used three different attribution methods here Integrated Gradients, Saliency and SmoothGrad.

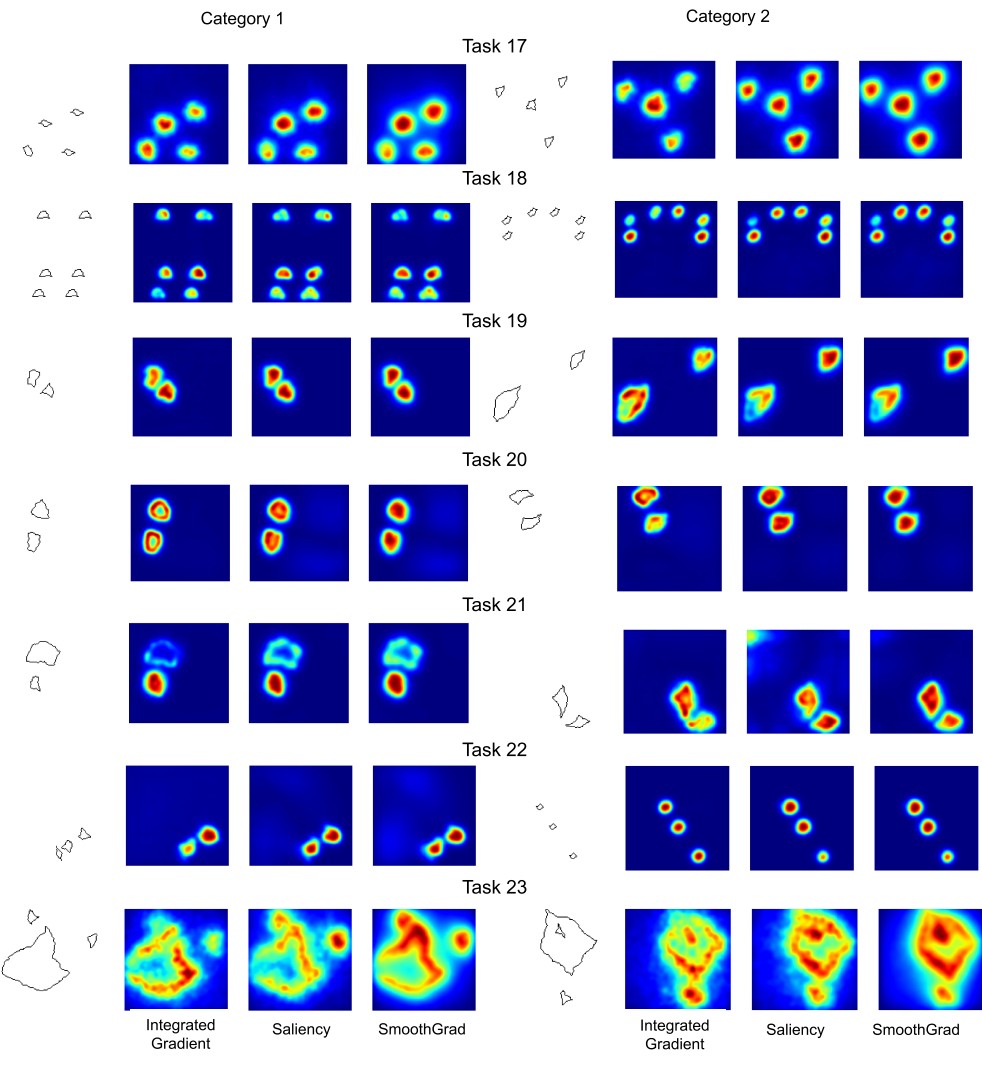

Figure S7: Attribution map visualization for SVRT tasks. The map represents the area of the image, which are significant in decision-making for *GAMR*. Tasks represented are from *17* to *23*. We used three different attribution methods here Integrated Gradients, Saliency and SmoothGrad.

## S5    EXPLANATION OF ROUTINES LEARNED AND CORRESPONDING ATTRIBUTION MAPS

This section explains the different mechanisms our model learns when we train it with a task and shows how it uses those learning mechanisms to understand the rule while testing on a novel set of tasks.

In Figure S8, we see the explainability maps corresponding to the model trained with tasks *5, 19, 20 & 21* and we test those models on task *1*. Figure S5 shows that in task *5*, the model learned a shortcut to select only one identical pair from a group of two pairs to take a decision. The model learns this shortcut because it was sufficient to make the correct decision. At the same time, to decide on images from category 2, the model stops searching as soon as it finds a non-identical region between two shapes.

In task *19*, the model learns scale invariance. The model chooses one shape and examines a similar part around the second shape to check the scale-invariant representation. A broader version of transformations is learned in task *21*, where the model understands rotation, scaling, and translation concurrently. So the model uniformly attends to the two shapes present in the image and compares them.

In task *20*, the model learns one of the three types of similarity transformation, i.e., reflection. If a model learns to distinguish similarity, it can solve any reasoning tasks associated with same-different identification.

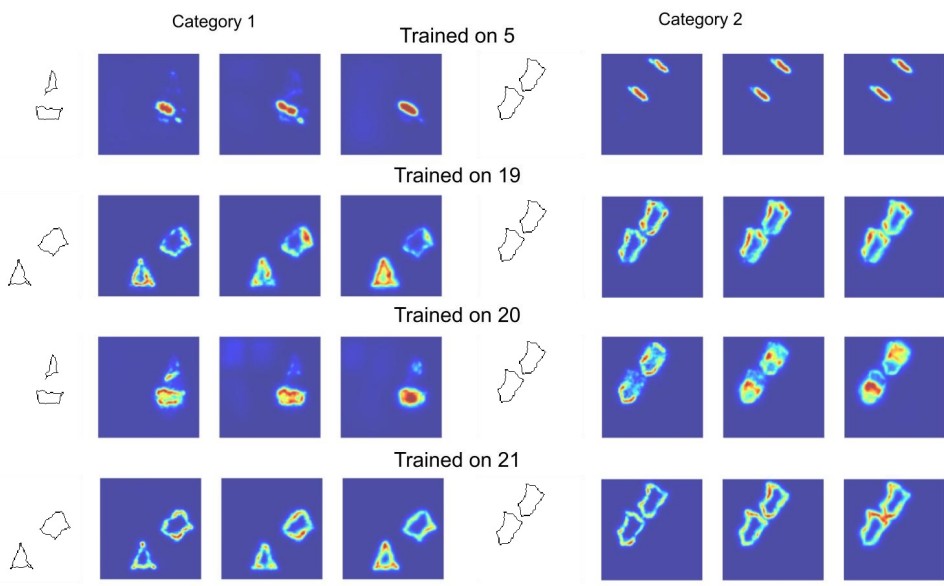

Figure S8: Attribution maps corresponding to the mechanisms involved in visual routines when testing on images from task *1* and the model is trained on different tasks as mentioned above each example image from both the category.

Moving on to Figure S9, we test the rules learned by tasks *1, 7 & 20* and test on task *22* where the challenge is to identify if the three shapes arranged in a row are identical or not. As seen from the previous discussions, task *1* learns to identify if two shapes are the same or not. As there are three shapes existing, the model tries to compare two shapes in pairs before deciding. So we can see the high heat map values on the two shapes for both tasks. As discussed before, task *20* learned a similarity transformation between two shapes, and it makes a pairwise comparison between three shapes to take a decision.

Glancing at the performance of task *7* on task *22*, we claim that the model learned to compare the shapes as well as count them to make a decision. Similar to category 2 of task *7*, here again, we have three shapes responsible for such a good performance.

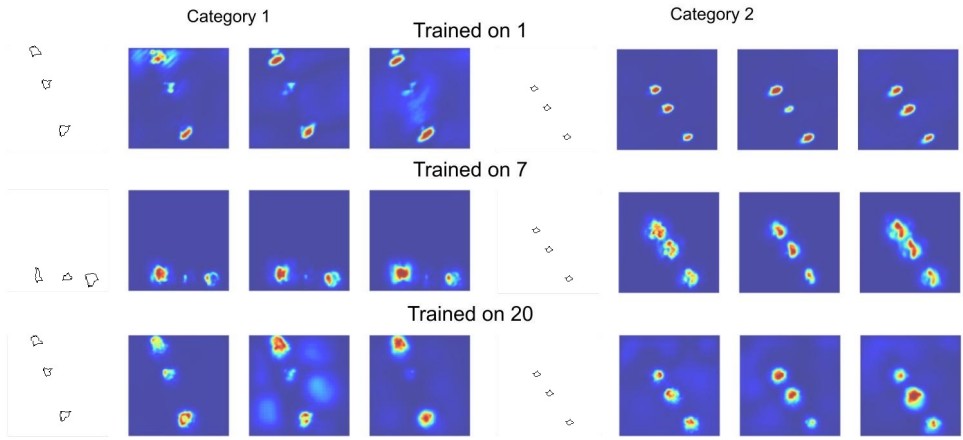

Figure S9: Attribution maps corresponding to the mechanisms involved in visual routines when testing on images from task *22*, and the model is trained on different tasks as mentioned above each example image from both the category.

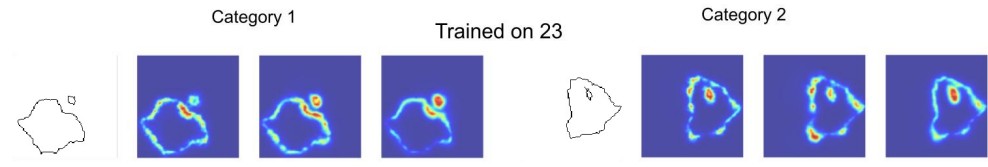

Figure S10: Attribution maps corresponding to the mechanisms involved when training a model on task *23* and testing on images from task *4*. One of the rules learned in task *23* is to see if a smaller shape exists inside the large shape, which is the same rule to be tested in category 2 of task *4*.

Next, we look at the strategy used to solve task *15* in Figure S11. The rule to formulate the task involved identifying if the four shapes required to form a square are identical. The model trained on task *1, 5, 21* and *22* starts by picking up one shape and stop comparing when it comes to the fourth shape. This is the reason all of the four shapes in category 2, which are similar, are seen to be involved in decision-making.

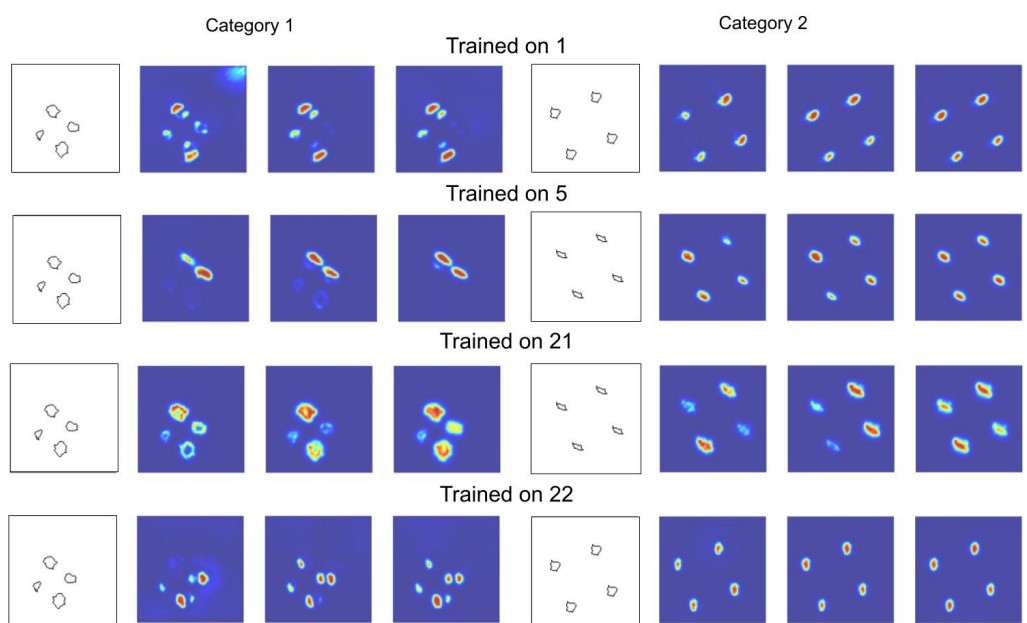

Figure S11: We show the visual routines learned by *GAMR* when training the model on task *1, 5, 21, 22* and we test the same model on task *15* (four shapes forming a square). It shows that once the model has learned to recognize one pair of identical shapes, it can extend the ability to count all the identical shapes present in an image. The visualization shows the attribution methods Integrated Gradients, Saliency, and SmoothGrad from left to right.

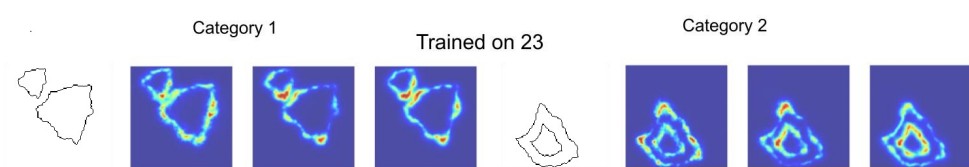

Figure S12: Attribution maps corresponding to the mechanisms learned when training a model on task *23* and testing on task *8*. One of the rules learned in task *23* is to comprehend if one small shape exists inside the large shape. A somewhat similar rule involved in task *8* is to identify if the smaller shape existing inside is also identical or not. The model is probably only answering on the bases of insideness without considering the same-different judgment.

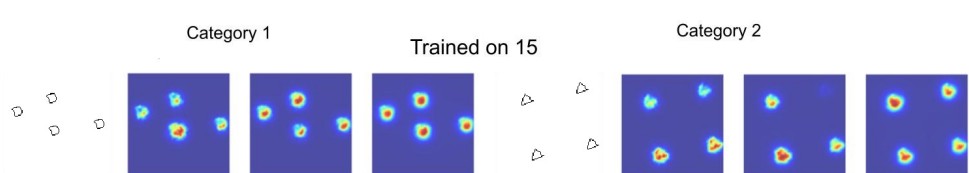

Figure S13: Attribution maps corresponding to the mechanisms learned when training the model on task *15* and testing on task *10*. Rules required in task *15* is to say if the four identical shapes are forming a square or not. Model, when trained on this task *15*, learns to form a square using four shapes in category 2, so it also solves the task *10*.

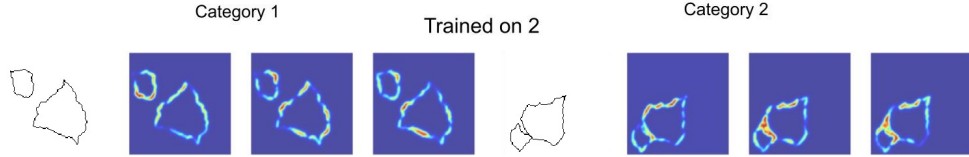

Figure S14: Visual routines learned when training the model when on task *2* is if the two shapes touch from inside or not. We test the model on task *11*, where the two shapes touch from the outside. It shows that the model has learned to recognize touch and use the same relation to solve the task.

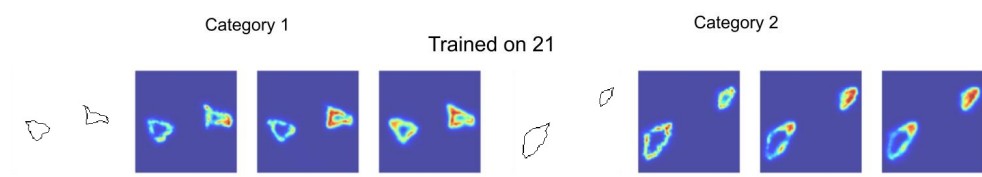

Figure S15: Task *21* involves rules related to the geometric transformation related to scaling and rotation. One of the components of this rule is present in task *19* related to scaling, which helps it to understand this task.

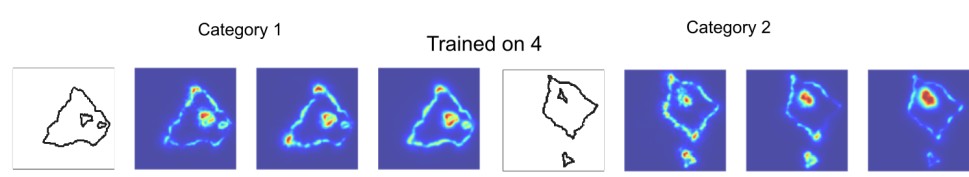

Figure S16: Attribution maps corresponding to the mechanisms learned while training the model on task *4* and testing on images from task *23*. In task *4*, the model learns the rules to say if one shape is present inside the other or not. This case is true for category 2 of task *23*. We expect the model to answer the same even for cases from category 1, where two smaller shapes are present inside the larger shape.

## S6  EXAMPLE SVRT TASKS

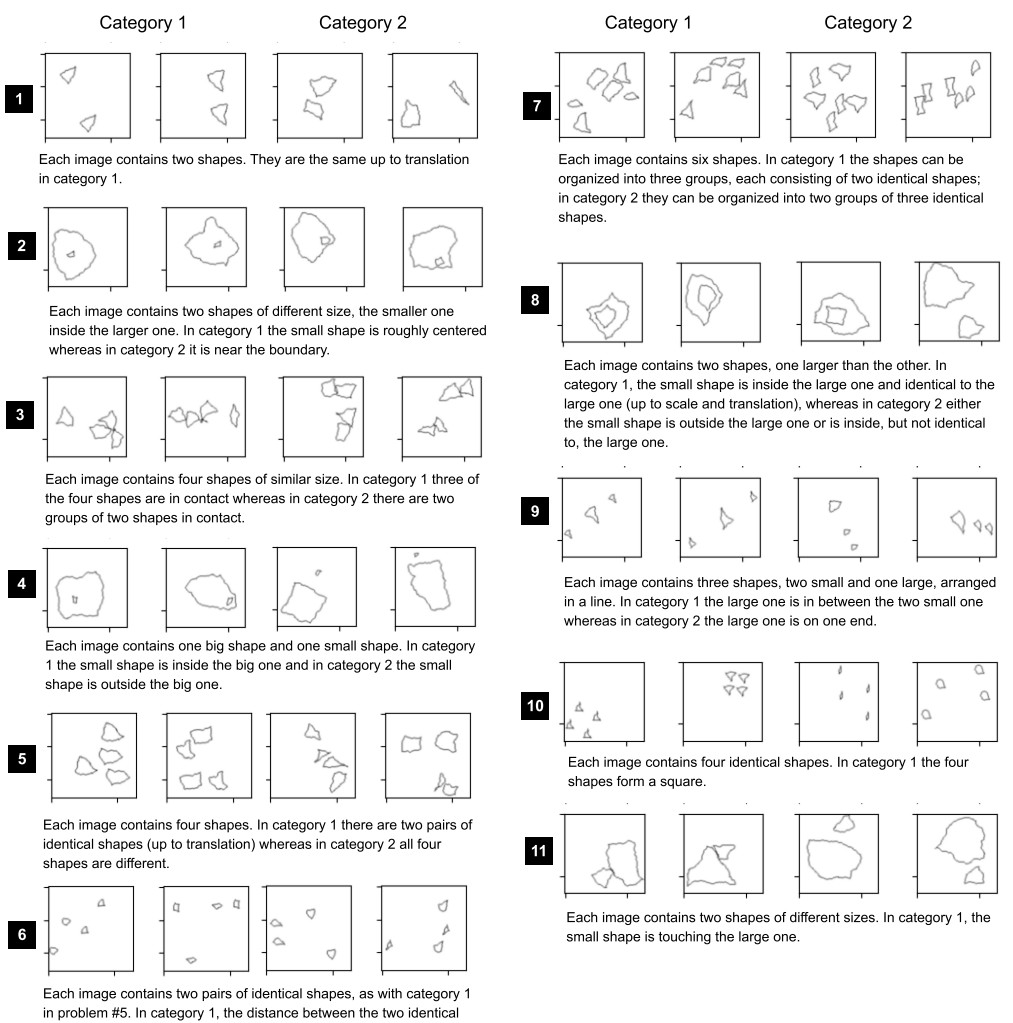

Figure S17: Two representative examples from each category of 23 SVRT tasks Fleuret et al. (2011). Each example has their tasks number labeled in the black box with its description mentioned below.

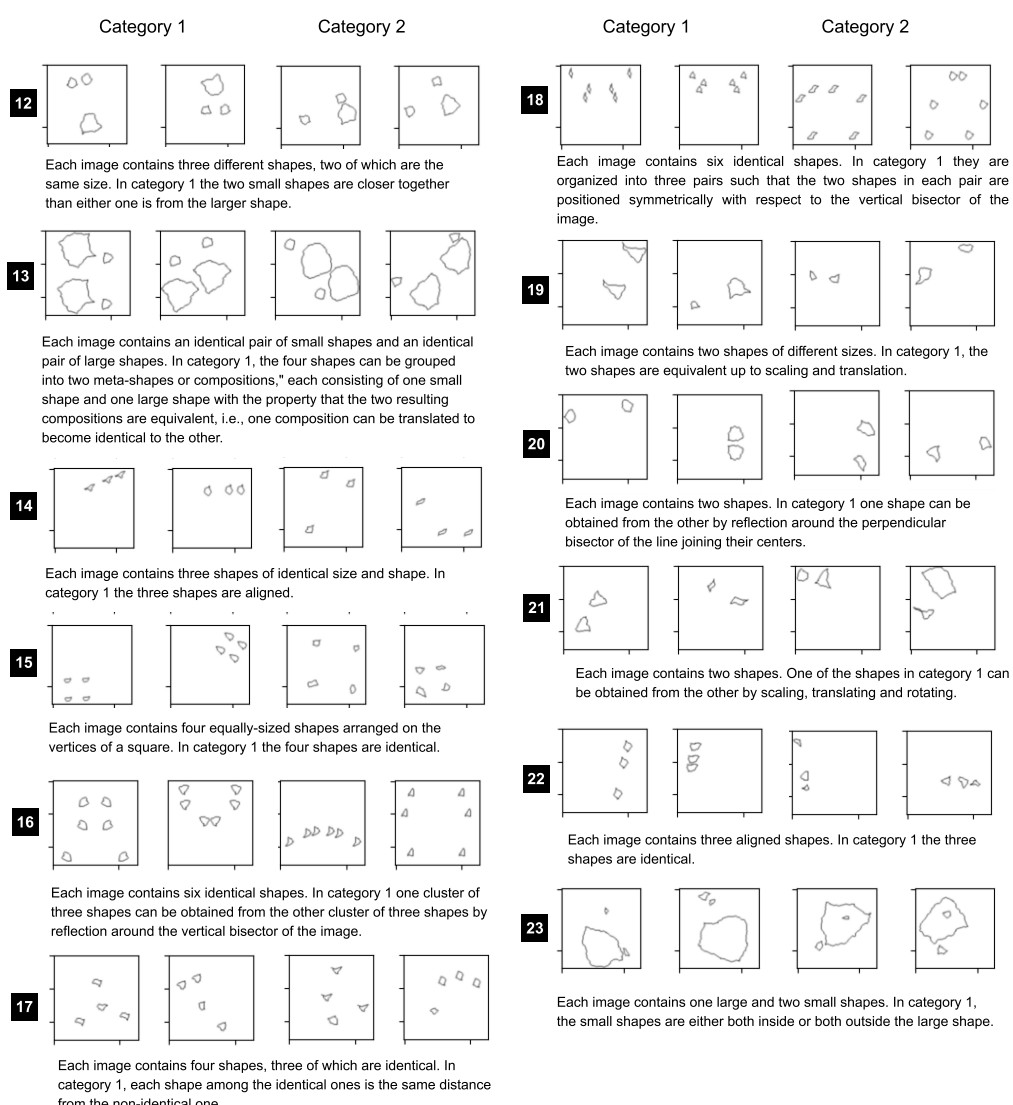

Figure S18: Two representative examples from each category of 23 SVRT tasks Fleuret et al. (2011). Each example has their tasks number labeled in the black box with its description mentioned below.

## S7 HYPERPARAMETERS

Table S7: **ART**: Number of training and testing samples used for four different types of tasks.

| Tasks | | m=0 | m=50 | m=85 | m=95 |
|---|---|---|---|---|---|
| SD | Training | 18,810 | 4,900 | 420 | 40 |
| | Test | 990 | 4,900 | 10,000 | 10,000 |
| RMTS | Training | 10,000 | 10,000 | 10,000 | 480 |
| Dist3 | Training | 10,000 | 10,000 | 10,000 | 360 |
| ID | Training | 10,000 | 10,000 | 10,000 | 8,640 |
| | Test | 10,000 | 10,000 | 10,000 | 10,000 |

**Holdout set** For example, holdout *0* represents a generalization regime in which the test sets contain the same characters as those used during training. At the other extreme, in holdout *95*, the training set contains a minimal number of characters, most of which are actually used for tests. Hence, it is necessary to learn the abstract rule in order to generalize to characters in this regime.

Table S7: **ART**: For four different tasks number of epochs and learning rates (LR) used to train different architectures.

| Tasks | m=0 | | m=50 | | m=85 | | m=95 | |
|---|---|---|---|---|---|---|---|---|
| | GAMR | | | | | | | |
| | Epoch | LR | Epoch | LR | Epoch | LR | Epoch | LR |
| SD | 50 | 0.0001 | 50 | 0.0005 | 100 | 0.0005 | 200 | 0.001 |
| RMTS | 50 | 0.00005 | 50 | 0.0001 | 50 | 0.0005 | 300 | 0.0005 |
| Dist3 | 50 | 0.00005 | 50 | 0.0001 | 50 | 0.00005 | 300 | 0.0005 |
| ID | 50 | 0.00005 | 50 | 0.00005 | 50 | 0.0005 | 100 | 0.0005 |
| | Other baselines | | | | | | | |
| SD | 50 | 0.0005 | 50 | 0.0005 | 100 | 0.0005 | 200 | 0.0005 |
| RMTS | 50 | 0.0005 | 50 | 0.0005 | 50 | 0.0005 | 300 | 0.0005 |
| Dist3 | 50 | 0.0005 | 50 | 0.0005 | 50 | 0.0005 | 300 | 0.0005 |
| ID | 50 | 0.0005 | 50 | 0.0005 | 50 | 0.0005 | 100 | 0.0005 |

**SVRT** This dataset can be generated with the code [2] provided by the SVRT authors with images of dimension $128 \times 128$. No augmentation technique was used for training other than normalization and randomly flipping the image horizontally or vertically, as is customary for this challenge (Vaishnav et al., 2022).

We trained the model for a maximum of 100 epochs with a stopping criterion of 99% accuracy on the validation set. The model was trained using Adam (Kingma & Ba, 2014) optimizer and a binary cross-entropy loss. All the models were trained from scratch. We used a hyperparameter optimization framework *Optuna* (Akiba et al., 2019) to get the best learning rates, and weight decays for these tasks and reports the test accuracy for the models which gave the best validation scores.

**Learning compositionality** One of the triplets (*21, 19, 25*) involves rotation which needed more than 1,000 samples to learn. So, we selected 5,000 samples each from the tasks $x$ and $y$ to pre-train the network. This pre-training is carried out for 100 epochs for both tasks. Once the model is trained, we fine-tune it on the novel unseen task $z$. We confirmed that *GAMR* was able to learn the new rule with as few as ten samples per category – hence demonstrating an ability to harness compositionality.

---

[2]https://fleuret.org/cgi-bin/gitweb/gitweb.cgi?p=svrt.git;a=summary

## S8  ESBN vs. GAMR

We have identified some limitations of the ESBN architecture, such as its sensitivity to translation and noise. It is also incapable in scenarios where multiple shapes are presented together in the stimulus or complex relations exist (Figure S19).

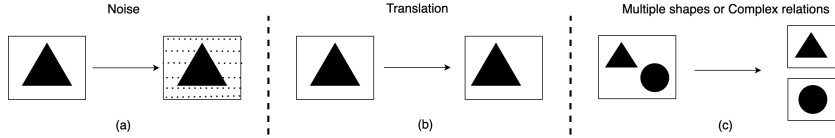

Figure S19: Some scenarios where the memory-based network, *ESBN* (Webb et al., 2021) performs at a chance level which shows the need for explicit attention mechanisms required in addition to memory to build a robust system. (a) Adding any kind of noise to the dataset, (b) If the image is not centered, (c) If there are multiple shapes (or the presence of complex relations like in the SVRT dataset).

To compare our architecture with that of ESBN, we analyzed *GAMR* on ART tasks and analyzed ESBN on SVRT tasks.

At first, we evaluated *GAMR* on the ART dataset as shown in Figure S20. In the previous work Webb et al. (2021), each image is passed as an individual image to the ESBN model, which is stored in the external memory bank for contemplating the rule, whereas *GAMR* is smart enough to figure out the rules from a single stimulus containing all the characters (Figure S21). This time we added translation to every character before passing. To take into account all these complexities introduced, like numerous shapes, relations, and stimuli altogether, we let our model run for 2 additional time steps (t=6) for tasks Dist3 and ID. For these two tasks, for our model to hold up the same level of performance as without translation, it was difficult without increasing the number of time steps. We compared the accuracy of our architecture and found that even in the hardest generalization scenario (m=95), it performs relatively well, where 95% of the shapes were held during training. Our model achieves at least 60% test accuracy when trained from scratch using ten different random seeds. The holdout of 95% is relatively hard for *GAMR* as it takes some time for it to develop an attention mechanism.

We have already seen in section 4 that ESBN architecture has difficulty contemplating the rules for the SD and RMTS tasks. To ensure that the updated backbone is not an issue, we ran a similar experiment where we used the same encoder module ($f_e$) as the one proposed in Webb et al. (2021) and obtained a consistent chance level accuracy.

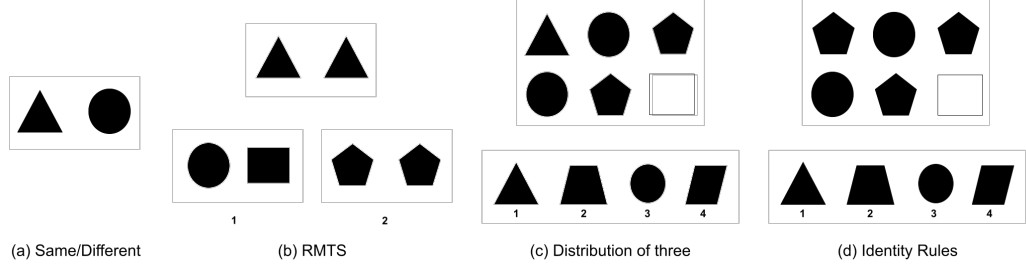

Figure S20: **ART:** (a) Same/different discrimination task. (b) Relational match-to-sample task (answer is 2). (c) Distribution-of-three task (answer is 1). (d) Identity rules task (ABA pattern, answer is 3).

As our next step, we moved on to the part where we analyzed the performance of ESBN on SVRT tasks. To make SVRT tasks compatible with the ESBN architecture, we disintegrated the shapes of the dataset in different image frames such that their location is intact. We did not center shapes in the

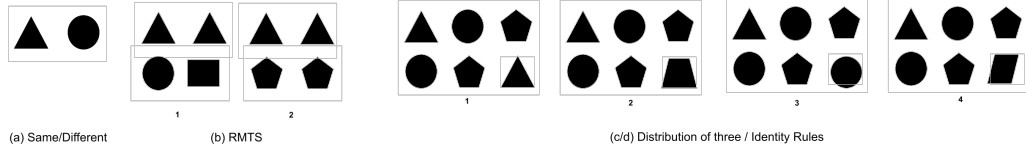

Figure S21: **ART for *GAMR*:** (a) Same/different discrimination task. (b) Relational match-to-sample task (answer is 2). (c) Distribution-of-three task (answer is 1). (d) Identity rules task (ABA pattern, answer is 3).

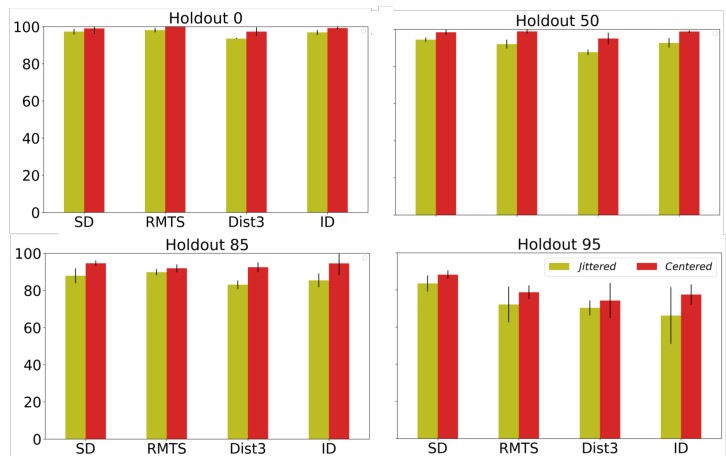

Figure S22: Test accuracy on ART with different holdout sets when the images are *centered* and compare the accuracy when shapes are *jittered* in every image. We find that unlike other baselines experiencing a huge drop in performance when shapes are jittered, GAMR is stable. We plot the average accuracy over ten runs on the dataset. $x$ axis corresponds to the four types of tasks, and $y$ represents the average accuracy score. These tasks are as follows: (a) same-different (SD) discrimination task, (b) Relation match to sample task (RMTS); (c) Distribution of three tasks (Dist3); and (d) Identity rule task (ID).

dataset as it might alter the rules imbibed in the task because of their spatial relational structure. A representative example of this method can be seen in Figure S23. Still, two of the tasks (*11* and *2*) in SVRT cannot be accustomed to this paradigm as they involve the concept of touching. So we skipped those tasks from our analysis because of the complex pre-processing steps involved in separating two touching contours into separate channels.

After this pre-processing step, we pass these individual images to the ESBN model and plot the performance in Figure S24. We found that ESBN even struggles with relatively simple tasks that were learned by ResNet-50 and other non-attentive models. Such tasks can be solved with approximately 500 samples, whereas ESBN did not learn to solve these tasks (*14, 9, 23, 10, 2*) even after training with 10k samples. We believe that the inability to formulate the spatial relations between shapes presented as individuated images is one of the significant drawbacks of architectures based on object-centric attention like ESBN.

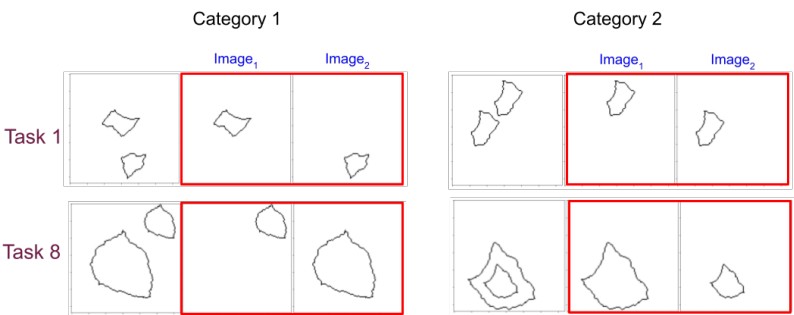

Figure S23: SVRT tasks compatible with ESBN model where each object is separated into individual images and passed to the ESBN in a sequential manner.

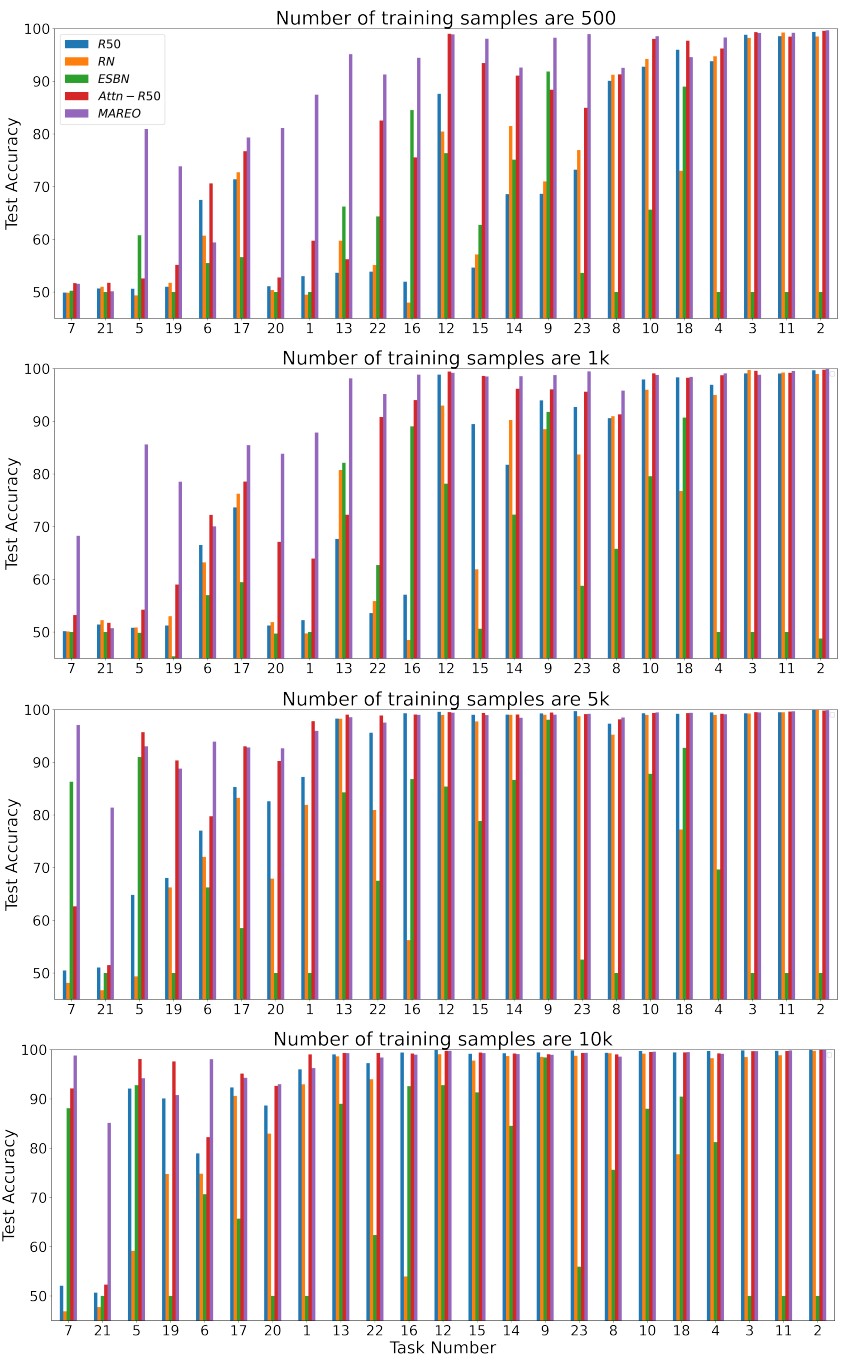

Figure S24: Comparing task level test accuracies of ResNet-50 ($ResNet$), Relation Network ($RN$), ResNet-50 with attention (*Attn-ResNet)* and *GAMR* for four training set sizes 500, 1k, 5k and 10k. Tasks are arranged as per the taxonomy introduced in Vaishnav et al. (2022) representing the rules involved in defining a task. For $ESBN$ model, task *3, 11* are not evaluated.

## S8.1 NOISE ROBUSTNESS

In this experiment, we tested the robustness of these two architectures against noise for two different datasets. We selected the most basic task of same-different discrimination (Figure S20) and used both the architecture in their naturalistic training paradigm. During training and testing, we added Gaussian noise to the images. We found that ESBN is not able to pass chance level accuracy even for the 0-holdout case where all the symbols are presented during the training time, whereas *GAMR* experienced an insignificant drop in accuracy (Figure S25). We observed a similar trend in the performance when we repeated the same experiment with a complex variant of same-different tasks, i.e., SVRT, and plotted the results in Figure S26. We matched the number of parameters in the encoder block ($f_e$) in both these experiments.

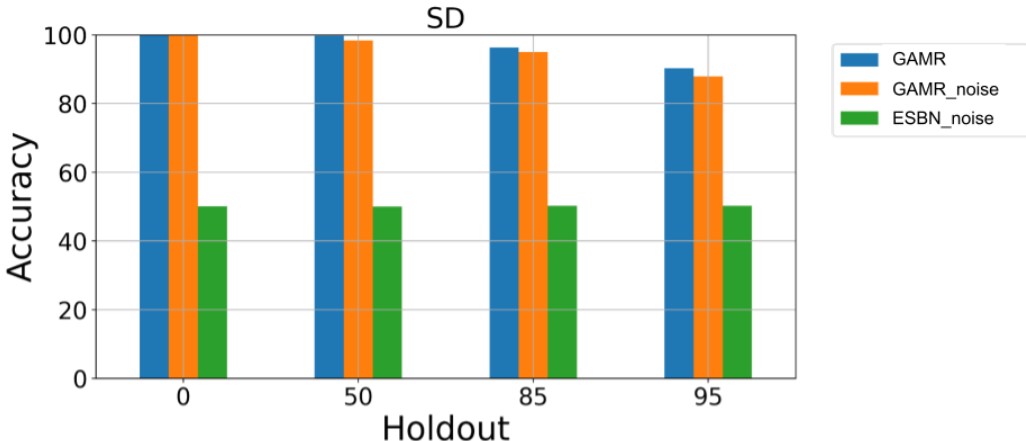

Figure S25: Test accuracy plot for same-different (SD) differentiation task when training and testing *GAMR* with the presence of Gaussian noise with different holdout values. For comparison, we also plot the model's accuracy without any noise and show an insignificant difference in accuracy between these two cases. However, ESBN struggles to learn at the 0 holdout scenario, thereby representing its inability to attend to shapes because it lacks an attention mechanism.

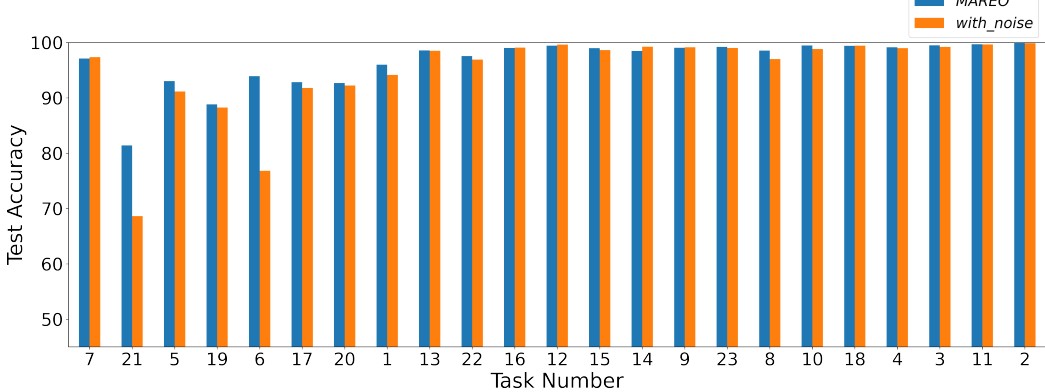

Figure S26: Test accuracy plot when training and testing *MARIO* with the presence of Gaussian noise using 5k samples on all twenty-three SVRT tasks. For comparison, we also plot the model's accuracy without any noise and show an insignificant difference in accuracy between these two cases.

