# OpenReview forum: "GAMR: A Guided Attention Model for (visual) Reasoning"
_ICLR.cc/2023/Conference — ICLR 2023 poster_

### Official Review · Reviewer_4tvR · 2022-10-16

**Confidence:** 4
**Clarity, Quality, Novelty And Reproducibility:** The paper is clearly presented with q…
**Correctness:** 4
**Technical Novelty And Significance:** 2
**Empirical Novelty And Significance:** 2
**Recommendation:** 6

**Strength And Weaknesses:**

The module design follows intuition. It's believed that memory and attention play a significant role in human cognition. The model is crafted following this theory and makes perfect sense. The experimental part is also very detailed: apart from checking in-distribution performance, the authors also verify compositionality and zero-shot generalization with illustrative charts. The model is performative compared to others, showing the superiority compared to the chosen baselines.

That being said, I do have some concerns for the paper.

For one thing, the model may not be very novel. Attention and memory have been extensively studied, either jointly or separately, in both vision and language. One very related work is Hahne et. al 2019. The model also shares very similar design to ESBN.

For another, the model is only evaluated on SVRT and ART, two very simple datasets. There have been more challenging ones for abstract reasoning, like PGM, RAVEN, Bongard, Bongard-HOI. It'd be better to show results on these datasets in order to show superiority. Besides, the authors only briefly mention Hahne et. al 2019 without comparing it extensively.

**Summary Of The Paper:**

The paper proposes a Transformer-based module for visual reasoning, the Guided Attention Module (GAMR). The model is instantiated with three components: an encoder module, a controller module, and a relational module. The encoder module uses CNN and TCN to extract image features. A guided attention module uses multi-head attention to route visual information from relevant time step and modulate the features to yield a new context vector. The controller LSTM module generates query for the guided attention. The relational module leverages a memory and the context vector to produce the final answer. In experiments, GAMR achieves new state-of-the-art on SVRT and ART. It also shows compositionality and zero-shot generalization.

**Summary Of The Review:**

I think the paper is in general in a good shape, but I'm concerned about novelty in model design and experimental comparison. I'd like to temporarily treat it as borderline and will sync with fellow reviewers.

---

> ### Author Response · Authors · 2022-11-12
> **Response to reviewer 4tvR**
>
> **Comparison with Hahne et. al 2019**:  The architecture referred to as GAMR self-attention (GAMR-SA) in the manuscript is a close approximation of the Hahne et. al 2019 ARNe architecture and we evaluated that model on SVRT tasks. In GAMR-SA we replaced the guided attention block of GAMR with a transformer self-attention (SA). GAMR-SA and ARNe both include a CNN encoder, followed by transformer-like self-attention layers and a set of MLPs to model relations. The accuracy of GAMR-SA on SVRT drops by 20% for same-different (SD) tasks and 8% for spatial-relation (SR) tasks (see Figure 3b). The somewhat narrower improvements on SR tasks compared to SD tasks simply reflect the fact that GAMR is already at the ceiling. We hope this clarifies the difference between the two architectures.
>
> **Regarding comparison with ESBN**: While ESBN was indeed a source of inspiration, we would like to emphasize that GAMR constitutes a substantial improvement over ESBN. First and foremost, ESBN lacks attention. It requires items/objects to be passed serially one by one and hence it cannot solve SVRT or any other multi-object visual reasoning problems. In a sense, the approach taken in ESBN  is to assume an idealized frontend that uses hard attention to perfectly parse a scene into individual objects and then serially pipe them through the architecture. This is where our work makes a substantial contribution by developing an attention front-end (which is soft and not hard) to sequentially attend to individual objects and route them into memory. We have updated the main text in Related work section.
>
> **Regarding performance on challenging tasks**: We agree with the reviewer that it would be interesting to test our architecture on more challenging tasks, but this would require a very different approach and we feel would be better kept for future work. There are two separate communities working on visual reasoning. Researchers in computer vision typically focus on applications of visual reasoning such as visual question answering (VQA). This combines multi-modal tools. Datasets such as RAVEN/PGM might contain multiple sources of biases. Researchers typically focus more on accuracy gains and less on cognitive science questions (e.g., what is the role of attention or memory, computational demand, etc.). Researchers in cognitive science like ourselves typically use synthetic datasets such as SVRT because they are well-controlled. The corresponding tasks can be framed as classification problems and architectures are typically tested on one of the two aforementioned datasets. The focus is less on accuracy and more so on understanding the role of certain brain mechanisms, including attention and memory.
>
> The RAVEN dataset is a saturated dataset [1-4] and ARNe is already achieving at, or near ceiling on it. Since GAMR outperforms ARNe, evaluating our model on these datasets does not seem necessary. On the other hand, though SVRT and ART have simple stimuli, network architectures such as ResNet50, ESBN, Transformer, and Relational Networks struggle to learn to solve them. However, we do plan to extend this architecture to make it compatible with a system similar to that of RAVEN/PGM where there are eight context images and eight choices.
>
> ```
> [1] Wu, Yuhuai, Honghua Dong, Roger Grosse, and Jimmy Ba. "The scattering compositional learner: Discovering objects, attributes, relationships in analogical reasoning." arXiv preprint arXiv:2007.04212 (2020).
>
> [2] Sahu, Pritish, Kalliopi Basioti, and Vladimir Pavlovic. "SAViR-T: Spatially Attentive Visual Reasoning with Transformers." arXiv preprint arXiv:2206.09265 (2022).
>
> [3] Hersche, Michael, Mustafa Zeqiri, Luca Benini, Abu Sebastian, and Abbas Rahimi. "A Neuro-vector-symbolic Architecture for Solving Raven's Progressive Matrices." arXiv preprint arXiv:2203.04571 (2022).
>
> [4] Hahne, Lukas, Timo Lüddecke, Florentin Wörgötter, and David Kappel. "Attention on abstract visual reasoning." arXiv preprint arXiv:1911.05990 (2019).
>
> ```

---

> > ### Comment · Reviewer_4tvR · 2022-12-08
> > **Reply**
> >
> > Thanks for the clarification. I still believe additional comparison on other datasets would better strengthen the points made in this work, but I'm OK with the current version.

---

### Official Review · Reviewer_SVeq · 2022-10-23

**Confidence:** 5
**Correctness:** 3
**Technical Novelty And Significance:** 4
**Empirical Novelty And Significance:** 4
**Recommendation:** 6

**Clarity, Quality, Novelty And Reproducibility:**

The paper is lacking in clarity in some places, and the organization of the paper could be improved.

In particular, the method is not described in sufficient detail. It is ok for this to be done in the supplementary information, but a complete description of the method needs to be present somewhere in the paper. In particular, the following things are missing or unclear:
- The primary component, guided attention (a novel part of the proposal), is not described in the algorithm, but instead is treated as if it were a widely familiar, predefined function, i.e. '*guided_attention()*'. This component should be explicitly described.
- Sums are described using a pseudo-pytorch syntax, rather than standard sum notation, and there is no explanation of which dimensions are being summed over. Ideally, the description of the method should be sufficiently detailed so as to be able to implement it without familiarity with pytorch conventions.
- The operator $\ast$ is not defined. I think this is elementwise multiplication, but this should be specified. $\odot$ is a more common notation.
- The LSTM $f_{s}$ generates a number of outputs, presumably through different output layers. This needs to be specified somewhere, along with the nonlinearities used in each layer.
- In the memory retrieval step, the memory matrix $M_{t-1}$ is multiplied by $w_{k_{t}}$, which I believe is a scalar, which is then multiplied (elementwise?) by $g$, a vector. How does this yield a vector? Is the result of the multiplication summed across the rows of the matrix?
- $r_{\theta}$ is described as a 'multilayer perceptron (MLP) layer', is this a single layer or an MLP? What are the hyperaparameters (nonlinearities and number of units)?
- The convolutional encoder is described as consisting of 'convolutional blocks', but it is not described what these blocks consist of. How many layers are in each block? What are the hyperaparameters?
- How many units are there in the LSTM hidden / cell state?
- In the code provided by the authors, the model appears to use dropout in some places, but this is not mentioned in the text.

Additionally, the organization of the paper is confusing in some places. First, the section entitled 'benchmarking guided attention' effectively describes an ablation study, that would make more sense to combine with the other ablation results later in the paper. Second, the paper partially describes both the SVRT and ART experiments in the 'Method' section, but then also presents some of the details on the ART experiments later in the section where the ART results are presented (called 'additional experiment'). I think it would be less confusing if the paper had *either* completely separate sections for SVRT and ART, in which the experimental details and results for both are contained, *or* one section that described all of the experimental details for both of these experiments, followed by the results section.

**Strength And Weaknesses:**

# Strengths:
- The proposed method integrates dynamic attention, memory, and relational processing, all of which are thought to be important components for human visual reasoning.
- The proposed method is able to perform visual reasoning over multi-object scenes.
- The proposed method shows reasonably strong out-of-distribution generalization, and can learn from relatively few examples.
- The experiments identify some interesting limitations of previous methods, such as the vulnerability of the ESBN to visual noise.

# Weaknesses:
- The controller module has a lot of complexity, and some of the components appear to be unnecessary. It would be easier to gain insight into how the model operates if these unnecessary components were removed.
    - I believe that it is not possible for the multi-head attention component to be contributing to the model's performance. In multi-head attention, for every query, an output is produced which is a weighted average of the values (weighted by the dot product between the queries and keys). However, in this case, there is only a single value, $query_{t}$ (this is also a confusing naming scheme), therefore the resulting outputs can only be copies of $query_{t}$. To confirm this, I used the code provided by the authors and ran a version of the proposed model that ablates the multi-head attention, on a few of the ART tasks. The results (averaged over 10 runs each) confirmed that the multi-head attention has no effect:
| | same/differenet | RMTS | Dist-3 |
| ----------- | ----------- | ----------- | ----------- |
| w/ MHA | $82.3\pm1.6$ | $64.6\pm3.7$ | $66.0\pm4.0$ |
| w/o MHA | $83.7\pm1.3$ | $67.3\pm2.7$ | $67.3\pm3.3$ |
This is also consistent with the results in the paper showing that the number of heads in the multi-head attention had no effect.
    - The results of the ablation study in the paper show that $w_{k_{t}}$ has no effect. Therefore, this component should be removed from the model.
- For the ART tasks, only GAMR performs the tasks using multi-object images as inputs (as opposed to receiving pre-segmented objected as the baselines do). Performing the task in this manner should ostensibly be more difficult, but it is also possible that there's some bias (such as visual entropy, see [1]) that can be exploited in this case that is not easily exploited when performing the task over segmented objects. The authors should compare to another baseline that performs these tasks using multi-object images as input, such as the ResNet used in the SVRT experiments.

[1] Joel Fagot, Edward A Wasserman, and Michael E Young. Discriminating the relation between relations: the role of entropy in abstract conceptualization by baboons (papio papio) and humans (homo sapiens). Journal of Experimental Psychology: Animal Behavior Processes, 27(4):316,
2001.


### Other concerns:
- The paper describes SVRT and ART as 'the two main visual reasoning challenges'. This is certainly not true, as there are many other such tasks, including PGM, RAVEN, relation games, CLEVR, CLEVRER, etc.
- In the 'related work', the PGM and RAVEN datasets are micharacterized. First, they are described as a single dataset called 'Raven's Progressive Matrices', but Raven's Progressive Matrices is a different problem set, on which the PGM and RAVEN datasets are (loosely) based. Second, these datasets are dismissed as being 'seriously flawed'. However, the papers cited to support this claim address a bias that is only present in the RAVEN dataset, not the PGM dataset, and one of these papers proposes an alternative version of this dataset, I-RAVEN, that addresses these concerns.
- I believe that the proposed model does not actually employ temporal context normalization, which normalizes representations across a temporal sequence, but instead uses *instance normalization* [2], which normalizes across the spatial dimension of a feature map.
- The 'GAMR w/ self-attention' and 'GAMR w/o attention' ablations should be described in more detail.
- The ESBN should be given a brief high-level description in the 'baselines' subsection.
- In the test of compositional generalization, what is the baseline? It is only described in the figure as 'baseline', and I couldn't find a description in the text.
- The evidence for compositional generalization could be stronger. How does GAMR perform if only trained on one of the source tasks, or if trained on unrelated source tasks?
- It would be helpful to give an illustration of one of the compositional tasks (both the two source tasks and the target task) in the main body of the paper.
- When describing the spatial jittering of objects in presenting the ART results, it should be clarified that this differs from the original implementation of these tasks, and that this explains the discrepancy between the results in the present work and the results in the original paper.

[2] Ulyanov, D., Vedaldi, A., & Lempitsky, V. (2016). Instance normalization: The missing ingredient for fast stylization. arXiv preprint arXiv:1607.08022.



**Summary Of The Paper:**

This paper proposes a novel method for visual reasoning, GAMR, that combines dynamic attention, memory, and relational reasoning. The method compares favorably against other popular approaches, and shows some evidence of compositional combination of rules.

**Summary Of The Review:**

In summary, I think the model presented in this work, and the results are promising, but a few extra control experiments are needed, and the clarity of the paper needs to be improved. Specifically, I think the following changes would be necessary to merit a higher score:
- A more careful ablation analysis, particularly looking at whether the multi-head attention component is needed. Unnecessary components (including also $w_{k_{t}}$) should be removed from the model.
- A control experiment needs to be performed on the ART tasks using a baseline that solves these tasks the same way GAMR does, using multi-object images as input (e.g. the ResNet baseline used in the SVRT experiments).
- The method needs to be described in more detail.

---

> ### Author Response · Authors · 2022-11-12
> **Response to reviewer SVeq**
>
> We are thankful to the reviewer for the in-depth analysis of the architecture and suggestions for better understanding.
>
> **MultiHead Attention (MHA)**: MHA is a characteristic feature of the self-attention module used in the Transformer network by Vaswani et al. and it gives the network greater expressiveness. We are grateful for the reviewer’s careful evaluation conducted on the ART task. It is consistent with our own analysis on SVRT (see SI; Figure S2 -- old manuscript) where we found insignificant but visible differences between the performance across different numbers of heads. We agree that these differences are not very significant and in the reasoning benchmarks used in this study, there is no obvious benefit of using multiple heads and a single-headed GAMR appears to be sufficient to solve all the tasks. We have updated the manuscript text and SI sub-section S7 Hyperparameters.
>
> **Placing $w_{k_t}$  in the final architecture** Our overall analysis shows that $w_{k_t}$ has no effect on performance when averaged over all the SVRT tasks. However, as shown in SI Figure S2 (Ablation studies for GAMR) when we broke down performance for each individual task it became apparent that more complex tasks (tasks 7) do indeed benefit from the  $w_{k_t}$ variable.
>
> **Control experiment for multi-object setup**: We are thankful for the reviewer’s suggestion to run an experiment to show that our model does not exploit other kinds of biases.  We ran the experiment suggested with a multi-object setup. We found ResNet50 which has more than 3 times the number of trainable parameters as GAMR consistently underperforms when trained in a similar setup to that of GAMR with multiple images embedded in a single stimulus. We have added this analysis to the main text (Figure 5 and also main text section 8 Baseline models).
>
> **Control experiment for compositionality**: This is yet another valuable suggestion from the reviewer. We followed the suggestion and trained our model on the composition of 2 tasks (5,17) and tested it on tasks 15, 18 and 21. The pair of tasks (5,17) does not include the rule needed to compose tasks 15, 18 and 21. So, the tasks (5,17) were unable to exploit any compositionality prior and performed at chance. We have updated the main text in section 5.
>
> **Clarity**: Thank you. We have followed the reviewer’s suggestions and updated the text to include all the proposed suggestions to make the paper and the pseudocode clearer.

---

> > ### Comment · Reviewer_SVeq · 2022-11-15
> > **Reply**
> >
> > Thanks to the authors for these thoughtful responses. Below is my point-by-point reply:
> >
> > ### MultiHead Attention (MHA):
> >
> > I thank the authors for adding a note to the supplementary about the impact of using multiple heads, but I don't think this adequately addresses the issue at hand. The key issue in the analysis that I presented is not whether multi-head attention adds anything above and beyond single-head attention, the key issue is whether the self attention is adding anything to the model at all (even with a single head). It is true that in general, self attention can give a network greater expressiveness, however, in this special case, in which there is only a single value vector (previously $query_{t}$, now $q_{int_{t}}$), *it is mathematically impossible for self attention to have any impact on the model's performance whatsoever*. Since the output of self attention is a set of weighted averages of the input value vectors, when there is only a single value vector, the only possible output is a set of copies of that value vector. Therefore, self attention cannot do anything in this case other than produce copies of the input ($q_{int_{t}}$), such that completely removing the box labelled 'Multihead attention' in Figure 1 will have no effect on the model's performance. I think it is unnecessarily confusing to include a component in the model that cannot possibly have any effect on its performance. It would be much easier for the reader to gain insight into the model if this was removed. Furthermore, there is currently not any mention of these issues in the manuscript, except for the small note in the supplementary material about the issue of multiple heads (which does not address the key issue of whether self attention has any effect whatsoever).
> >
> > ### Placing $w_{k}$ in the final architecture:
> >
> > Thanks to the authors for adding this more detailed analysis. How many runs were performed for each condition in the figure? There are no error bars, so it is possible that the differences merely reflect differences in random initialization, rather than having anything to do with the presence of $w_{k}$ per se.
> >
> > ### Control experiment for multi-object setup:
> >
> > The results of this analysis do indeed seem to show that the multi-object version of this task is, surprisingly, easier than the version presented to the ESBN, RN, and transformer. It is true that GAMR outperforms the ResNet baseline; however, the ResNet baseline performs very well -- it is competitive with the ESBN and outperforms the RN and transformer. That does suggest that presenting the tasks in this manner somehow makes them easier, even if GAMR still retains an edge. I think some note should be added to the text addressing this issue.
> >
> > When presenting the task in a multi-object format, is the random spatial jitter applied independently to each object, or is the same randomly sampled jitter applied to all objects in a given problem?
> >
> > ### Control experiment for compositionality:
> >
> > Thanks to the authors for adding this control experiment, it is very informative, and stregnthens the original conclusions re: compositionality.
> >
> > ### Clarity:
> >
> > Thanks for addressing these issues, the clarity of the paper is now substantially improved. There are however a few remaining issues:
> > - PGM and RAVEN are still mischaracterized in the 'Related work'. First, these datasets are still referred to as 'Raven's Progressive Matrices', which is a distinct pre-existing problem set. Second, there is no mention of the fact that I-RAVEN was introduced to address the bias identified in RAVEN, nor any mention of the fact that no such bias has been identified in PGM. The impression given to the reader is that the bias identified in RAVEN has effectively undermined the utility of all of these datasets, which is incorrect.
> > - The addition of the blue text specifying the dimensionality of all variables in the algorithm statement really helps to clarify things. However, it is missing from the step that produces $k_{r_{t}}$. It is still unclear to me how this step works. It involves the multiplication of a scalar with a matrix (which should produce another matrix), followed by elementwise multiplication with a vector. Is this vector tiled so as to match the dimensionality of $w_{k_{t}} \cdot M_{t-1}$? Is the result of the elementwise multiplication ($k_{r_{t}}$) a vector or a matrix? If a matrix, how is this then passed as input to the LSTM controller on the next timestep, given that the matrix will grow on every timestep?

---

> > > ### Author Response · Authors · 2022-11-17
> > > **Response to additional comments**
> > >
> > > **Multihead Attention Module** We appreciate the reviewer’s efforts for the timely explanation so that we could conduct further experiments. We thought about the implementation and ran an experiment removing the self-attention block but letting the residual connection and LayerNorm. In this setup, we consider key, query, and value as $z_{img}$, $q_{int_t}$, and value $z_{img}$. At first, we added the query to the key which is a form of additive attention. This form of attention is used to augment the features in the $z_{img}$ corresponding to $q_{int_t}$. Next, we pass the attention-augmented vector through a LayerNorm and take a mean across the spatial dimension to obtain $w_k$. This vector ($w_k$) is multiplied with the $z_{img}$. This modification simplified the controller module in terms of understanding while maintaining the performance of GAMR. For evaluation, we trained the model with 500 and 1k samples. We found that the performance on 23 SVRT tasks in both setups is similar. However, when we tested the zero-shot generalization ability of the network following a similar procedure as described in the paper, we found significant differences between the performances of the two models. So we think that self-attention is playing an important role in enabling GAMR generalizability ability. We have included this point in the ablation study and added the following Table in the SI.
> > >
> > > | Training | Test | GAMR  | GAMR_Additive |
> > > |----------|------|-------|---------------|
> > > |          | 5    | 72.07 |    62.26         |
> > > | 1        | 15   | 92.53 |    63.23         |
> > > |          | 22   | 84.91 |    57.20         |
> > > |          | 1    | 92.64 |    95.44         |
> > > | 5        | 15   | 84.36 |    88.56         |
> > > |          | 22   | 76.47 |    74.46         |
> > > | 7        | 22   |  83.8 |    83.96         |
> > > | 21       | 15   | 90.53 |    84.28         |
> > > | 23       | 8    |  85.84 |    51.94     |
> > >
> > > **Role of $w_{k_t}$** We do now believe that $w_{k_t}$ is not essential in the final model. Error bars across different runs showed that this component can be removed. We have updated the text accordingly.
> > >
> > > **Control experiment for multi-object setup** We have updated the Results in Section 8 discussing the possible benefits of a multi-object setup. We applied random spatial jittering independently to each shape in a multi-object setup. However, the performance differences in a stronger generalization regime (holdout=95) and also the performance gain of GAMR over other baselines in multiple experiments on SVRT tasks where all the shapes are presented in the same manner show its superiority.
> > >
> > > **Clarity** We believe that both RAVEN and PGM are Raven’s Progressive Matrices style tasks, hence we referred to both while mentioning RPM tasks. To avoid confusion, we have now removed the reference to PGM in the updated text. We have also updated the Algorithm to show that for $M_{t-1}$ (which was missed in the previous update) we sum over the time steps and take the element-wise multiplication with $g$.

---

> > > > ### Comment · Reviewer_SVeq · 2022-11-17
> > > > **reply**
> > > >
> > > > ### Multihead attention:
> > > >
> > > > I remain unconvinced that it is possible for self-attention to be playing any role in the proposed architecture. I have the same concern regarding these new results as I had previously regarding the ablation of $w_{k_{t}}$: how many runs was this ablation analysis performed over, and how variable were the results? How were these particular task combinations selected, and were there other task combinations evaluated where the difference went the other direction? The basic concern is that it's not at all clear whether these differences reflect anything other than differences in random initialization.
> > > >
> > > > I would appreciate if the authors could more directly address my conceptual point regarding the redundancy of self-attention in this case. Self-attention produces, for each query vector, a weighted average of the value vectors. If there is only a single value vector, as is the case here, it is impossible for the output to be anything other than a copy of that value vector. Is this incorrect? If not, how is it possible for self-attention to produce anything other than a copy of the inputs in this case?
> > > >
> > > > The authors note that they found that the self-attention module has no impact on all 23 SVRT tasks. But I cannot find this result in the revised manuscript, nor is there any mention of the fact that self-attention has no impact in the ART tasks. Thus, the manuscript does not currently provide an accurate picture of the role that self-attention plays in this architecture.
> > > >
> > > > ### Role of $w_{k_{t}}$:
> > > >
> > > > I thank the authors for addressing this point.
> > > >
> > > > ### Control experiment for multi-object setup:
> > > >
> > > > I thank the authors for adding a note addressing this issue. I agree that the results do nevertheless show an advantage for GAMR above and beyond the advantage conferred by using a multi-object setup.
> > > >
> > > > ### PGM and RAVEN datasets:
> > > >
> > > > I still think the text still presents a misleading picture regarding these datasets. First, PGM is clearly related work, and just as relevant as RAVEN, so needs to be mentioned. Second, the text states that RAVEN has been 'revised as I-RAVEN', but does not make it clear that this revision *addresses the previous flaws in RAVEN*, such that it is not possible, with I-RAVEN, for neural architectures to solve the task by leveraging shortcuts, as the authors suggest.

---

> > > > > ### Author Response · Authors · 2022-11-18
> > > > > **Reply**
> > > > >
> > > > > We went to the bottom of the issue and found that we both are correct here. We looked at the PyTorch implementation of the MultiHead Attention module. Interestingly, their implementation consisted of several combinations of the linear layer across the model which transformed the “v” ($q_{int_t}$) vector. There are also projection weights and biases for the attention matrix and which is why the attention matrix is not a copy of $q_{int_t}$. With the use of multi-head attention, $q_{int_t}$ is broken down into four separate 32-D vectors which are concatenated back to form a transformed $q_{int_t}$. We do now see the differences between what the reviewer meant and what we obtained when we ran the code. We have added this point to the discussion.
> > > > >
> > > > > We also want to share that in this work, we have trained more than 1000 models for the baselines and GAMR. So we preferred to conduct them on a fixed seed (12345) unless required to optimize for the available compute.
> > > > >
> > > > > We really appreciate the reviewer’s keen interest in improving the work and hope that all the other reviewers will receive the final version of the paper of the work in a constructive manner.

---

> > > > > > ### Comment · Reviewer_SVeq · 2022-11-18
> > > > > > **Reply**
> > > > > >
> > > > > > Thanks very much to the authors for digging into this issue further. This sounds like a good explanation for how different results are obtained from the same seed both with and without multihead attention. However, it does mean that, in this case, what is labelled 'multihead attention' in the model architecture is no more than a linear layer. Splitting up the output of a linear layer and then re-concatenating it is equivalent to simply taking the output of the linear layer. This explains why there are no differences overall on either SVRT or ART when ablating the multihead attention altogether. When there is only a single value input, there is nothing for attention to operate over (other than producing a copy of that value). Therefore, the current model would be more parsimoniously described by simply removing the 'multihead attention' component from the description of the architecture, since it is in reality only an additional linear layer applied to $q_{int_{t}}$.
> > > > > >
> > > > > > I take the authors' point about not being able to perform multiple runs for each baseline and ablation, given the very large number of variations. Using a fixed seed is a reasonable approach, but it does mean that caution needs to be taken in interpreting the results. One cannot conclude that, given that the same seed produces different results on two different variants of a model, those same differences would robustly emerge given different random seeds. There will always be idiosyncratic interactions between a given seed and a specific set of hyperparameters.
> > > > > >
> > > > > > I want to emphasize that I have an overall positive impression of this work, but I would be much more strongly in support of acceptance if the issues regarding the architecture were clarified. I think it will be much easier for readers to gain insight into how the model works if the redundant 'multihead attention' component is removed and described instead as a linear layer (which is probably not an important component of the model, and thus could be treated more as an implementational detail). At the very least, the paper needs to make some mention of the fact that removing the multihead attention has no impact overall on either SVRT or ART. But much simpler would be not to mention multihead attention at all.

---

> > > > > > > ### Author Response · Authors · 2022-11-18
> > > > > > > **Reply**
> > > > > > >
> > > > > > >  We thank the reviewer again for their healthy discussion and we assure to incorporate these suggestions in the final manuscript.

---

### Official Review · Reviewer_o1fP · 2022-10-24

**Confidence:** 4
**Correctness:** 4
**Technical Novelty And Significance:** 3
**Empirical Novelty And Significance:** 3
**Recommendation:** 8

**Clarity, Quality, Novelty And Reproducibility:**

# HIgh-level remarks:

- The paper is generally well-written and of high quality. Some details could be clarified/improved, but I am confident that the authors can make these improvements during the rebuttal period.
- The paper is reproducible. The main body and appendix provide an appropriate level of information, and the authors release the code for the work. The code is nicely documented and well organized.


# Detailed questions/suggestions:

- One question for the authors: I could not find how the ResNet data from Figure 2 was computed. Is that computation part of the supplementary information zip file?

- “Inspired b active vision theories” should be “Inspired by active vision theories”

- The font size in Figure 1 is very small. At the same time, there is wasted whitespace to the left and right of the figure. I recommend that the authors use the whitespace on the sides of the figure, and increase the font size.

- “and selectively route” should be “and selectively routes”

- The capitalization in “REasOning” (in Algorithm 1) is odd. Was this intentional? Maybe GAMR used to have a different acronym, in which the capitalization made more sense?

- In Figure 1, a dotted red line indicates information flow of the “out” state into the reasoning module. In other places, data flows are indicated using solid arrows. Why not the same for the “out” state? Or did I miss something here ?

- “t=T”  on page 3 should be in math mode.

- Even though papers referenced by the submission describe a memory bank, I recommend that the authors describe in a few sentences how the memory bank works. A reader may for example wonder if the memory bank contains only the current z_t, or all previous z_t too?

- The authors write “We trained the model for a maximum of 100 epochs with a stopping criterion of 99% accuracy on the validation set.” . This seems like an unusual stopping criterion. Why not just train until the validation loss no longer goes down for a few epochs (“early stopping”)?

- In Figure 2, the authors show performance with 500 training examples and more. The proposed GAMR architecture appears to perform at a very high level even with 500 training examples. Why did the authors not include a graph showing the performance with say 100 training examples? The performance with very few training examples is, in my opinion, the most exciting aspect of this research area.

- In the first paragraph of Section 6, the authors write “The first layer learns the …, and the second layer …” . Are these the layers of the MLP? It would be helpful if the authors could be more explicit.

- “and only fine-tuned” should be “and only fine-tune”.

- In Figure 3, what is the model labeled “Baseline”?

- It would be helpful if the authors included a figure showing some example challenges from the ART and SVRT datasets in the main body of the paper, or at least pointed out that examples are in the Appendix. While the authors cite the original sources, the ART and SVRT datasets are so central to the paper that readers benefit from not having to look up the sources.

- Section 7 is onerous to read because tasks that could better have been described in graphics were instead described in words. Please include some graphics illustrating the multi-shot scenarios in Section 7.

- “t=4” should be in math mode.


**Strength And Weaknesses:**

# Strengths

- The proposed method’s architecture is quite elegant, and it works well empirically.
- The proposed method’s effectiveness provides further evidence that memory and a main “controller module” in the brain are essential for reasoning.
- The paper is generally well-written.
- The paper’s supplementary information and appendix are great.
- The paper’s ablation studies are very nice.

# Weaknesses

- The authors could have used even harder visual reasoning tasks (which might have meant creating a new dataset). As the authors point out, the model currently uses 4 time steps only for its reasoning. Reasoning tasks harder than those from ART and SVRT may require many more time steps. It would be interesting to see how the proposed method performs in these scenarios.

**Summary Of The Paper:**

The authors present GAMR, a brain-inspired architecture for visual reasoning. This architecture is transformer-based, but additionally utilizes an LSTM controller module and a memory module. The architecture empirically works well on visual reasoning task. This is exciting from an engineering perspective, but also of interest to (computational) neuroscientists investigating the role of memory and attention (a central controller module) in (visual) reasoning.

**Summary Of The Review:**

This is a nice paper.

The proposed architecture is interesting both from an ML and engineering perspective (interesting from a conceptual perspective, and much improved performance over earlier methods) and from a neuroscientific perspective (further evidence that brain may need both memory and a central attention module to perform reasoning.

The proposed method’s architecture is quite elegant, and it works well empirically.

The paper’s supplementary information and appendix are great, and the paper’s ablation studies are very nice.

---

> ### Author Response · Authors · 2022-11-12
> **Response to reviewer o1fP**
>
> We thank the reviewer for the detailed comments and suggestions along with observant corrections.
>
> **Task requiring more time steps**: We have discussed the tasks requiring additional time steps in SI section S8 (updating it in the main text as well: section 8, parag. 2). The modified ART tasks with the introduction of jittering to the shapes complexified them requiring two additional time steps to achieve the state of the art performance.
>
> **Memory bank (M)**: You are right that the previous z_t is also stored in the memory bank (M). We have added this information in section 2, parag. 2 (it was otherwise shown in the pseudocode.)
>
> **ResNet50 data**: For ResNet50 and Attn-ResNet50, we followed a similar scheme as used in Vaishnav et al. 2022 (updated in section 3, Baselines, parag. 2). We trained both of these models for 100 epochs and 3 random initializations and chose the model with the best validation accuracy to evaluate on the test dataset.
>
> ```
> Mohit Vaishnav, Remi Cadene, Andrea Alamia, Drew Linsley, Rufin VanRullen, Thomas Serre; Understanding the Computational Demands Underlying Visual Reasoning. Neural Comput 2022; 34 (5): 1075–1099. doi: https://doi.org/10.1162/neco_a_01485
> ```
> **Stopping criterion**  We would like to correct the typo in our sentence that saturation in the validation accuracy is used as an **early** stopping criterion as done in Vaishnav et al. 2022. (updated the main text in section 3). We do not expect this to change the results as the validation loss is nearly zero when the validation accuracy is >=99% and remains there thereafter.
>
> **Training with 100 samples** When we trained the model with 500 samples, most of the baseline architectures performed at chance level, so we considered it redundant to further decrease the number of training samples.
>
> **MLP layer in Relational Module** It is a two-layer MLP (updated main text in section 2, parag. 3). We have also added SI section S1 detailing the implementation. (also pointed out by Reviewer SVeq).
>
> **Baseline model in Figure 3**  Please see our answer to  reviewer Gp6M. We have updated the main text in section 5.
>
> **Figures for SVRT and ART dataset**  Because of space constraints, we were unable to move the representative examples in the main text however we now ask the reader to refer to the figures in SI in the Dataset parag. of SVRT and ART experiment sections.
>
> **Figure for tasks in Zero shot Generalization**: Here again, we chose to describe the SVRT tasks in words and refer the reader to the SI for visual examples of those tasks to save space. In Figures S17 and S18, we show 4 representative examples of tasks along with their descriptions.

---

> > ### Comment · Reviewer_o1fP · 2022-11-29
> > **Thank you for the clarifications**
> >
> > Thank you for the clarifications!

---

### Official Review · Reviewer_Gp6M · 2022-10-25

**Confidence:** 3
**Correctness:** 3
**Technical Novelty And Significance:** 2
**Empirical Novelty And Significance:** 3
**Recommendation:** 5

**Clarity, Quality, Novelty And Reproducibility:**

The paper is unclear in parts, mainly in its justifications for design features and justifications for some of the baselines used. However I think the work is probably reproducible.

**Strength And Weaknesses:**

*Strengths:*

The paper provides some evidence that sequential visual attention in some form (although superficially different from how it works in humans) is important for visual spatial reasoning.
Comprehensive ablations clarified the important components of the model for performance.
The GAMR model seems to generalise to tasks with similar relations better than their baseline models like Attn-ResNet.

*Weaknesses:*

It is not clear to me how the task is presented to the model. The attentional processing is said to be task-dependent but Figure 1 shows no task-based inputs to the model.
It seems slightly strange to consider an attentional visual reasoning network which is said to be motivated by human visual reasoning, and have the system process the entire image at each time step. Humans and other primates instead take sequential saccades - they must learn a saccadic attentional policy (most similar here to what the LSTM is doing) but which directs this attention spatially to intake and process only small local portions of the visual scene at each timestep (that which fits within a foveated glimpse). This has led to other models which are more closely aligned with the human visual system, like the 2014 paper Recurrent Models of Visual Attention (Mnih 2014) which learns a saccadic policy over visual images. I am unsure whether results have been reported on the Mnih et al network’s performance on SVRT and ART datasets, however the architectural differences and differences in learned policy might be enlightening and help to establish deeper cognitive theories as a result of the work in this paper.

The current architecture is more comparable to a system which has its eyes firmly fixed at the centre of the image, and has to process it without using saccades to view and attend to parts of interest. This is wholly unlike how humans or animal view and attend to images (see papers measuring humans’ saccadic traces over visual scenes for more details). As a result the attentional maps that the GAMR model learns show distributed attention at each time step rather than local attentional processing. This is fine if the model is not intended to be a cognitive model of human visual reasoning, but the parallel to human visual processing is made at several points in the paper and thus feels incomplete. If the authors wish to keep the link to cognitive science, the paper could benefit from exploring and justifying these architectural choices more thoroughly.

The paper claims that GAMR is more computationally efficient than with SA because it yields higher performance for the same number of training samples (1k). But this does not consider the computational cost of a forward pass which is higher since there are multiple sequential sweeps through the guided attention module.
The paper claims that what the model learns is compositional - in that it re-composes previously learned sub-policies. The evidence for this was a little unclear as it’s difficult to interpret what the appropriate baseline is in Figure 3.

*Minor points:*

In introduction and justification of the work:
A single recent EEG study suggesting attention and memory are involved in same-different visual reasoning seems out of place here and a poor justification compared to the entire history of work on visual reasoning in cognitive science and neuroscience.
Points to ‘modern cognitive theories’ which are from the 1980s (not so modern).
The legend for the architecture diagram seems to be incorrect: The legend says that the recurrent controller f_s generates the queries at each time step, but these are input as keys and values into the transformer module while the queries seem to be sources from the encoded image.

It would be helpful to explain more clearly exactly how where the guided attn model and the self-attn models deviate - I am assuming that the LSTM is removed and z_img is used as query, keys and values but this is not stated explicitly. It is similarly unclear in this comparison whether there is some self-attn model replacement to make up for the ‘out’ channel into the reasoning module which would presumably be missing.


**Summary Of The Paper:**

This paper presents a transformer-based model for visual reasoning. The paper puts forward an architecture that shifts attention sequentially over a preprocessed image and outputs binary labels pertaining to the relationships between objects in the visual scene. The authors suggest that this model supports cognitive theories that attention and memory interact to solve complex visual reasoning tasks, which seems like quite an uncontroversial claim for the cognitive sciences.

The model preprocesses an image using a convolutional module, and then learns a task-dependent sequential attention policy over these preprocessed features by using recurrence from an LSTM to direct attention queries over multiple sequential passes through a transformer.

The model performs well on spatially-related (SR) tasks from the SVRT dataset compared to baseline models in the low data regime (on datasets smaller than 1k), and similarly to baselines when more data is used. In same-different tasks the GAMR model outperforms the baselines in all experiments, with the main benefits again noticed in the low data regime. On the Abstract Reasoning tasks dataset GAMR again outperforms baselines.

The major contribution of this work is to highlight the importance of recurrent attentional processing for solving visual reasoning tasks. As the model fairly consistently outperforms the presented baselines, this signifies a useful contribution to the literature. The ablations were thorough. The treatment of the related cognitive science feels a little superficial however, and would benefit from a more thorough treatment (or removal altogether - which would also be fine). The paper claims that what the model learns is compositional - in that it re-composes previously learned sub-policies.


**Summary Of The Review:**


This paper presents a transformer-based model that performs sequential attention over visual images. The results highlight the importance of attentional mechanisms in visual reasoning tasks. The results do not seem particularly novel to me, but I am not an expert on this particular body of literature. The paper itself would benefit from a more considered treatment of the relation to cognitive science and improvements in clarity throughout.

---

> ### Author Response · Authors · 2022-11-12
> **Response to reviewer Gp6M**
>
> We thank the reviewer for their valuable comments.
>
> **Regarding how tasks are presented to the model**: SVRT contains 23 unique tasks. Here, we followed the standard SVRT protocol and trained one model per task as a binary classification task.
>
> **Comparison with human visual reasoning**: There seems to be a bit of confusion regarding overt vs. covert attention but our model is completely plausible. There is evidence for overt attention shifts (i.e., associated with saccades) and covert (i.e., without saccades). Both are closely related and indeed overt attention is needed to be deployed at a location before a saccade can be made at that location (see premotor theory of attention [1]). From a computational standpoint, the only real difference between overt and covert attention exists for vision systems endowed with a fovea (i.e., a region of greater visual acuity at the center of the retina).The Mnih et al system constitutes an example of overt attention (and indeed the authors take inputs at a greater resolution at attended locations). Conversely, our system constitutes an example of a covert attention system and hence assumes a fixed acuity. Hence, the success of our approach provides evidence for the benefits of attention mechanisms per se and is not confounded by improvements that could arise because of greater acuity. It is also important to note that according to the premotor theory of attention covert attention is required prior to overt shifts/saccades. Hence it should be possible to extend our model in a biologically plausible way by explicitly enabling saccades to overtly attended regions and by taking into account different acuities at the fovea vs periphery. We have added this point to the SI subsection S2.1).
>
> ```
> [1] http://www.scholarpedia.org/article/Premotor_theory_of_attention
> ```
>
> **Comparison with Mnih’s model**: We thank the reviewer for suggesting this relevant architecture. To verify the claims, we ran the Mnih et al. model for the easier SVRT tasks. We found that the network is unable to learn task 2 requiring to find whether the smaller one of the two shapes is in the center of the larger one or around the boundary. The results are shown below. We used an image resolution of 60x60 and train the model for 200 epochs with 500 samples. We also trained the model with increased image resolution up to 128x128 but this led to a very significant drop in accuracy to near chance level.
>
> In addition, we would like to point out that the architecture proposed by Mnih et al. is non-differentiable and it is trained using reinforcement learning to learn policies, which makes it significantly different from our proposed end-to-end differentiable model. In short, we believe that there exist both qualitative and quantitative differences between the two classes of architectures and while the work by Mnih et al was pioneering it is far less expressive than our proposed GAMR. We have added a reference to Mnih et al and added the comparison to the SI subsection S2.1.
>
> | Accuracy | |glimpse  |  |  |
> |----------|---------|-------|-------|-------|
> |          | 4       | 5     | 6     | 10    |
> | Best validation acc | 78.00   | 72.00 | 74.00 | 72.00 |
> | Test acc     | 62.26   | 61.55 | 71.21 | 58.98 |
>
>
> **Computational efficiency of GA vs SA**: We have updated the main text to clarify that both self-attention (SA) and guided-attention (GA) models were trained for an equal number of recurrence steps (Section 7, parag. 1 and also see details in SI: Implementation details). Our experiments thus unambiguously demonstrate the benefits of the guided-attention GAMR over self-attention -- both computationally and quantitatively.
>
> **Regarding Compositionality**: We thank the reviewer for pointing out that the baseline is not described in the compositionality experiment (see also comment by reviewer o1fP, SVeq). As our baseline, we trained the model from scratch on task $z$ from the triplet of tasks (x, y, z) to show that the model is indeed exploiting compositionality. We have updated the text in section 5.
>
> **Minor comments**: Thank you for the suggestions. We have updated the manuscript accordingly with the recent references discussing the interplay between attention and memory.

---

> > ### Comment · Reviewer_Gp6M · 2022-11-20
> > **Thanks for the extra baseline work!**
> >
> > Thanks to the authors for their comprehensive response and for running additional experiments. Their point on the distinctions between overt and covert attention is fair, but this should be made explicit and discussed in the paper. The paper makes claims about attention generally, yet as the authors point out there are nuances here in how that attention manifests. This has ramifications for the connections to cognitive science - if the model is intended as a general model of visual attention and reasoning, the paper should at least discuss that a major feature of visual attention (overt saccades) is not considered here. If the connection between the model and biological attention is only via covert rather than overt attention then this should be made explicit in the text and addressed as a limitation.
> > Thanks for simulating Mnih et al on the same task - I think this comparison is helpful and is a nice empirical point for discussing the above distinction.

---

> > > ### Author Response · Authors · 2022-11-22
> > > **Reply**
> > >
> > > We thank the reviewer for their suggestions for improving the paper and bringing clarity. We will update the final version of the paper accordingly.

---

### Author Response · Authors · 2022-11-12
**General comments**

We thank all the reviewers for their time and effort in reviewing our work. We are pleased to see that reviewers have recognized the significance of the proposed model. They have agreed on the importance of incorporating attention and memory in ANNs as we demonstrate that these mechanisms significantly improve their reasoning ability. Reviewers have also noted the out-of-distribution and zero-shot generalization capabilities of the resulting architecture. Furthermore, they appreciated the systematic ablation study we conducted to demonstrate the contributions of individual circuit components to the overall architecture.

Although the reviewers appreciated the contributions of this work, they expressed concerns regarding the presentation and clarity. They asked for clarifications and made a number of suggestions for improvement including clarifying the architecture’s cognitive science underpinning, conducting an evaluation of the model on more complex tasks, providing a better justification for the number of heads used in the Multi-Head Attention block as well as specific comparisons to the ESBN architecture and Recurrent Attention Model (RAM) of Mnih et al.

We have revised the paper in a way that we think satisfactorily addresses all of the main points raised by the reviewers. We believe the manuscript has significantly improved as a result and we are grateful to the reviewers for their feedback. In particular, we have updated the manuscript per the reviewers’ recommendations and have run experiments to strengthen our claims.

---

### Decision · Program_Chairs · 2023-01-20

**Decision:**

Accept: poster

**Justification For Why Not Higher Score:**

Nice work in sequential attention for visual spatial reasoning but still preliminary

**Justification For Why Not Lower Score:**

This work will be of interest to, besides ICLR researchers, computational neuroscientists investigating the role of memory and attention for visual reasoning.

**Metareview: Summary, Strengths And Weaknesses:**

The paper investigates the role of memory and attention (a central controller module) in visual spatial reasoning with recurrent attentional processing. The authors present an architecture that shifts attention sequentially over a pre-processed image and outputs binary labels pertaining to the relationships between objects in the visual scene. The results shown generalise to tasks with similar relations better than their baseline models like Attn-ResNet, and also exhibit reasonable out-of-distribution generalization with learning from relatively few examples.

**Note From Pc:**

if the above contains the word "oral" or "spotlight" please see: "oral" presentation means -> notable-top-5% and "spotlight" means -> notable-top-25%. As stated in our emails, we are disassociating presentation type from AC recommendations

**Summary Of Ac-Reviewer Meeting:**

NA as 3 of 4 reviewers accept the paper. The 4th reviewer (score 5) is mainly concerned about clarity in presentation and comparison, which the authors have provided good responses